# THD-BAR: Topology Hierarchical Derived Brain Autoregressive Modeling for EEG Generic Representations

**Wenchao Yang**[1]* **Weidong Yan**[1]* **Wenkang Liu**[1]* **Yulan Ma**[1] **Yang Li**[1,2,3,4]†

[1]School of Automation Science and Electrical Engineering, Beihang University
[2]the State Key Laboratory of Virtual Reality Technology and Systems, Beihang University
[3]7T Magnetic Resonance Imaging Translational Medical Center, Department of Radiology, Southwest Hospital, Army Medical University (Third Military Medical University)
[4]Advanced Institute of Information Technology, Peking University

## Abstract

Large-scale pre-trained models hold significant potential for learning universal EEG representations. However, most existing methods, particularly autoregressive (AR) frameworks, primarily rely on straightforward temporal sequencing of multi-channel EEG data, which fails to capture the rich physiological characteristics inherent to EEG signals. Moreover, their time-centered modeling approach also limits the effective representation of the dynamic spatial topology of brain activity. To address these challenges and fully exploit the potential of large-scale EEG models, we propose a novel Topology Hierarchical Derived Brain Autoregressive Modeling (THD-BAR) for EEG generic representations. The core innovation of THD-BAR lies in the introduction of the Brain Topology Hierarchy (BTH), which establishes a multi-scale spatial order for EEG channels. This hierarchical structure enables a redefinition of autoregressive learning as a "next-scale-time prediction" problem, effectively capturing both spatial and temporal dynamics. Based on BTH, we design a Topology-Hierarchical Vector Quantized-Variational Autoencoder (THVQ-VAE) for multi-scale tokenization and develop an enhanced Brain Autoregressive (BAR) module with specialized masking strategies for prediction. Through extensive large-scale pre-training on 17 datasets, followed by rigorous validation on 10 downstream datasets spanning 5 distinct tasks, THD-BAR consistently outperforms existing methods. These results highlight the superior generalization and modeling capabilities of our proposed approach. Our code is available at `https://github.com/thdbar/THD-BAR`.

## 1 Introduction

Electroencephalography (EEG) is a fundamental tool in Brain-Computer Interfaces (BCIs) due to its non-invasive nature, relatively low cost, and high temporal resolution [1]. EEG-based BCI technologies demonstrate significant potential across diverse applications, including emotion recognition, motor imagery classification, mental workload assessment, sleep stage classification, and epilepsy detection. However, developing large-scale models that can effectively generalize across different tasks and datasets remains a significant challenge.

---

*These authors contributed equally.
†Corresponding author. Email: `liyang@buaa.edu.cn`

39th Conference on Neural Information Processing Systems (NeurIPS 2025).

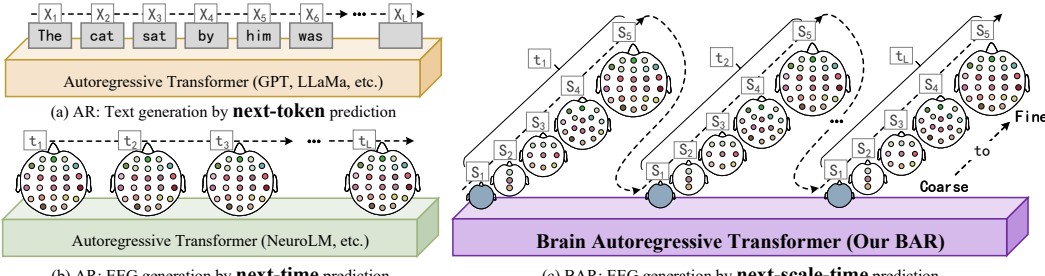

Figure 1: Conceptual comparison of autoregressive prediction strategies.

Recent advancements in artificial intelligence have opened new opportunities for EEG analysis. Specifically, motivated by the transformative success of large-scale pre-training techniques in natural language processing (e.g., autoregressive (AR) frameworks like GPT [2]) and computer vision (e.g., MAE [3]), the field is increasingly adopting similar strategies for EEG analysis. Initial efforts focused on tackling superficial data heterogeneity, such as inconsistent electrode configurations and sampling rates. Models such as MMM [4], LaBraM [5], and NeuroLM [6] among others [7–13] employed techniques like masked autoencoding, advanced tokenization, and AR architectures to improve robustness to these variations. While these advancements, particularly the adoption of AR frameworks [6, 8, 9], have improved robustness to superficial data heterogeneity, they encounter a more fundamental challenge: defining a sequence order for EEG that truly reflects brain dynamics. Current AR models typically emulate language processing by arranging multi-channel EEG data temporally and performing "next-time prediction" (Figure 1 (b)). However, this temporal-centric ordering struggles to adequately capture the inherent variability of brain activity. Neural activation patterns and their corresponding spatial topology on the scalp dynamically shift depending on the underlying cognitive task (e.g., processing visual stimuli versus experiencing emotions) [14]. This task-dependent spatial variability represents a deeper layer of heterogeneity that a purely temporal sequence fails to effectively address. Consequently, even sophisticated AR models are limited in their ability to model fundamental shifts in spatial brain dynamics, hindering the development of truly universal and generalizable EEG representations.

Given these limitations of current AR approaches, which struggle to model dynamic spatial topologies due to their predominantly temporal order assumption, a fundamental rethinking is necessary. This raises a crucial question: *How can we define the spatial order for EEG signals?* Inspired by hierarchical information processing in human perception and vision modeling (e.g., VAR [15]), we propose a novel concept called the Brain Topology Hierarchy (BTH), which establishes a nested "whole brain - brain region - channel" relationship grounded in physiological structure, thereby defining our proposed spatial order. Based on this hierarchy, we redefine AR learning for EEG as "next-scale-time prediction" (Figure 1 (c)), and design a novel Topology Hierarchical Derived Brain Autoregressive Modeling (THD-BAR) framework, facilitating the simultaneous modeling of dependencies across both spatial scales and temporal sequences. The THD-BAR framework is realized through several key components working in synergy: First, the BTH itself provides the foundational multi-scale mapping hierarchy, structuring the spatial context for EEG analysis. Second, to effectively capture the deep hierarchical features of EEG signals, we introduce the Topology-Hierarchical Vector Quantized-Variational Autoencoder (THVQ-VAE), which tokenizes EEG signals into discrete hierarchical representations by incorporating a modified multi-scale quantization layer into the standard VQ-VAE. Third, these multi-scale tokens generated by THVQ-VAE are then flattened according to a defined spatio-temporal order (hierarchically in space, then sequentially in time), preparing them for the autoregressive model. Fourth, the "next-scale-time prediction" strategy is performed by our Brain Autoregressive (BAR) module, which is an enhanced AR architecture integrating both scale-wise and time-wise masking. To evaluate the effectiveness of THD-BAR, we pre-train on a large-scale dataset comprising 17 EEG tasks to learn universal representations. We then assess its generalization by fine-tuning on 10 downstream datasets covering 5 major EEG applications. Experimental results show that our proposed method significantly outperforms existing approaches across diverse EEG tasks. Our contributions are listed below:

- We propose THD-BAR framework, a generic foundation model for EEG generic representation learning. Pre-trained on 17 diverse datasets, it captures complex spatio-temporal dynamics, yielding significant performance on various downstream tasks over existing methods.

- We introduce the BTH, which establishes a "whole brain - brain region - channel" relationship grounded in physiological structure. Building on this hierarchy, we develop the THVQ-VAE to generate discrete, multi-scale quantized tokens.
- A nested "next-scale-time prediction" strategy is employed for pre-training. This strategy compels the BAR module to learn complex spatio-temporal dependencies by predicting tokens hierarchically across scales within each time step before progressing to the next time step, thus modeling both intra-time hierarchical relationships and inter-time dynamics.

## 2 Method

In this section, we detail the comprehensive framework of THD-BAR. We begin by constructing a BTH based on the location of channels and the function of brain regions. Following this, the model development involves three key stages (as depicted in Figure 2): (1) training an THVQ-VAE as a specialized neural tokenizer to generate tokens; (2) pre-training our BAR model using these tokens with a novel autoregressive strategy; (3) fine-tuning the pre-trained BAR model on diverse downstream EEG tasks.

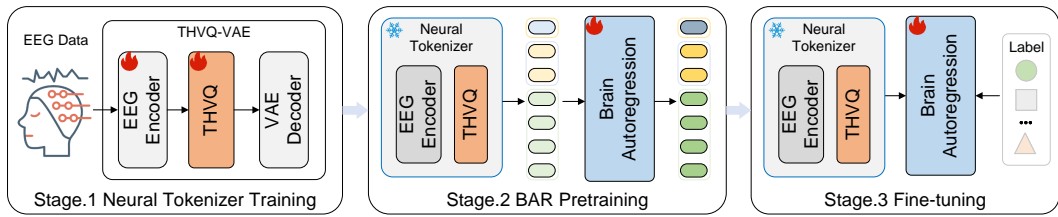

Figure 2: The three-stage pipeline of the THD-BAR framework. (1) An THVQ-VAE tokenizes unlabeled EEG data. (2) The BAR model undergoes autoregressive pre-training using these tokens. (3) The pre-trained BAR model is fine-tuned for specific downstream tasks with labeled data.

### 2.1 Brain Topology Hierarchy

Electroencephalography (EEG) signals inherently capture brain activity across multiple spatial scales. However, the non-uniform distribution of EEG electrodes poses a challenge in defining a structured hierarchy for analysis. To address this, we propose the Brain Topology Hierarchy (BTH), a framework that imposes a multi-scale spatial order on EEG channels, enabling a more nuanced modeling of brain activity. The BTH is conceptually constructed by referencing standard electrode placement systems (e.g., the 10-10 system) and grouping channels based on principles such as spatial proximity and their association with underlying brain regions [16]. This process establishes a series of progressively finer topological scales, ranging from a holistic representation of the entire brain scalp at the coarsest level, through intermediate groupings corresponding to broad brain regions, down to near-individual channel resolution at its most granular stages [14]. An exemplary implementation, utilized in our experiments, employs a specific number of distinct scales (e.g., five scales, S1 through S5, progressing from whole-brain to individual channels). A detailed description of the specific five-scale configuration used in our work, including the rationale for channel groupings at each level, is provided in Appendix D. Within this hierarchical structure, we denote the channel grouping at a specific scale $s$ as $ch_s$. Specifically, $ch_S$ refers to the finest scale, representing the original individual channel resolution. By establishing this BTH, we provide a structured framework for analyzing the spatial distribution of brain activity. This hierarchical organization is crucial for our model to learn dependencies and representations not just across time, but also across different physiologically relevant spatial scales, from global patterns to fine-grained local activity.

### 2.2 Neural Tokenizer Training

To transform continuous EEG signals into a sequence of discrete, hierarchically structured tokens, we train our proposed THVQ-VAE as a specialized neural tokenizer. The training process begins with preparing the input EEG data: Given arbitrary EEG signals $EEG \in \mathbb{R}^{N \times L}$, where $N$ represents the number of channels and $L$ denotes the length of the signal, we consider an EEG sample as $EEG_{sample} \in \mathbb{R}^{N \times W}$, where $W$ is the window size. This leads to a total of $\lfloor \frac{L}{W} \rfloor$ samples for each

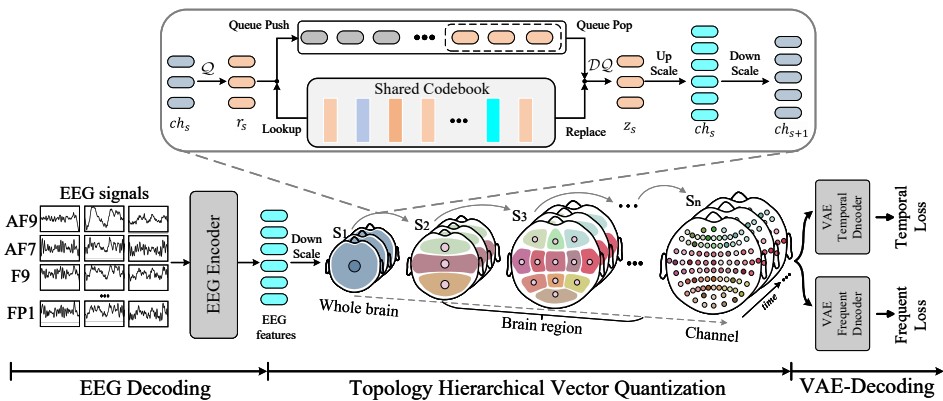

Figure 3: Topology-Hierarchical Vector Quantized-Variational Autoencoder (THVQ-VAE).

signal segment. Then, we segment the EEG samples along temporal domain into non-overlapping patches $eeg \in \mathbb{R}^{N \times T \times P}$, where $P$ represents the patch size, and $T$ is the number of patches, calculated as $T = \lfloor \frac{W}{P} \rfloor$.

**Preliminary: Vector Quantized-Variational Autoencoder.** A vector quantized-variational autoencoder (VQ-VAE), as described in works such as [5, 6], is commonly applied to encode EEG signal $eeg \in \mathbb{R}^{N \times T \times P}$ into EEG feature maps $f \in \mathbb{R}^{N \times T \times C}$, where $C$ is the dimension of the learned feature representation for each patch. These feature maps $f$ are then quantized into discrete tokens $q \in [V]^{N \times T}$, dequantized discrete tokens $q$ back into quantized feature maps $\hat{f} \in \mathbb{R}^{N \times T \times C}$, and finally decode the quantized feature maps $\hat{f}$ into reconstructed EEG signal $e\hat{e}g \in \mathbb{R}^{N \times T \times P}$. The complete process is as follows:

$$f = \mathcal{E}(eeg), \quad q = \mathcal{Q}(f), \quad \hat{f} = \mathcal{DQ}(Z, q), \quad e\hat{e}g = \mathcal{D}(\hat{f}), \tag{1}$$

where $\mathcal{E}(\cdot)$ denotes an encoder, $\mathcal{Q}(\cdot)$ a quantizer, $\mathcal{DQ}(\cdot)$ a dequantizer and $\mathcal{D}(\cdot)$ a decoder. Both the quantizer and dequantizer typically share a learnable codebook $Z \in \mathbb{R}^{V \times C}$, which contains $V$ code vectors. The quantization process $q = \mathcal{Q}(f)$ will map each feature vector $f^{(n,t)}$ to the code index $q^{(n,t)}$ of its nearest code in the Euclidean sense, and dequantization process $\hat{f} = \mathcal{DQ}(Z, q)$ will lookup quantization feature vector $\hat{f}$ of the code index $q^{(n,t)}$ from codebook $Z$:

$$q^{(n,t)} = \arg\min_{\nu \in [V]} \left\| \text{lookup}(Z, \nu) - f^{(n,t)} \right\|_2, \quad \hat{f}^{(n,t)} = \text{lookup}(Z, q^{(n,t)}), \tag{2}$$

where $\text{lookup}(Z, \nu)$ refers to selecting the $\nu$-th vector from codebook $Z$. Following the approach in [6], we predict both the original signals and the frequency magnitude of EEG signals. The Discrete Fourier Transformer (DFT) is applied to transformer each EEG patch $eeg^{(n,t)}$ into their corresponding frequency patch $fre^{(n,t)}$, using Euler's formula. A frequency decoder is employed to decode reconstructed frequency patch $\hat{fre}^{(n,t)} = \mathcal{D}(\hat{f})$. Additionally, as in [6], we introduce a domain classifier $C$ to predict whether the embeddings are from EEG or text. Finally, a compound loss $\mathcal{L}$ is minimized:

$$\mathcal{L} = \|eeg - e\hat{e}g\|_2^2 + \left\| fre - \hat{fre} \right\|_2^2 + \left\| f - \hat{f} \right\|_2^2 + \lambda \sum_i d_i \log C(f), \tag{3}$$

where $d_i$ denotes the label from either the EEG or text domain, while $\lambda = \frac{2}{1+e^{-10Step/Steps}} - 1$ is a scaling factor that smoothly transitions from 0 to 1 as the step progresses.

**Topology-Hierarchical Vector Quantized-Variational Autoencoder.** As shown in Figure 3, we design a THVQ-VAE to encode EEG into multi-scale discrete token maps $R = (r_1, r_2, \ldots, r_S)$, which are essential for our BAR. Building upon [6], our architecture incorporates modified multi-scale vector quantization and dequantization. These modifications leverage *downscale* operations to aggregate features from finer to coarser scales during quantization, and *upscale* operations to distribute features from coarser to finer scales during dequantization. These multi-scale procedures, which include residual designs on $f$ or $\hat{f}$ with $S$ extra convolution layers $\{\phi_s\}_{s=1}^{S}$ applied to the upscaled quantized feature vectors $z_s$ (at the finest scale $ch_S$) for refinement, are detailed in Algorithm 1 and Algorithm 2. A shared codebook $Z$ is employed across all scales to ensure that each $r_s$'s tokens belong to the same vocabulary $[V]$. Once fully trained, the autoencoder $\{\mathcal{E}, \mathcal{Q}, \mathcal{DQ}, \mathcal{D}\}$ is frozen to tokenize EEG data for subsequent unidirectional autoregressive model training.

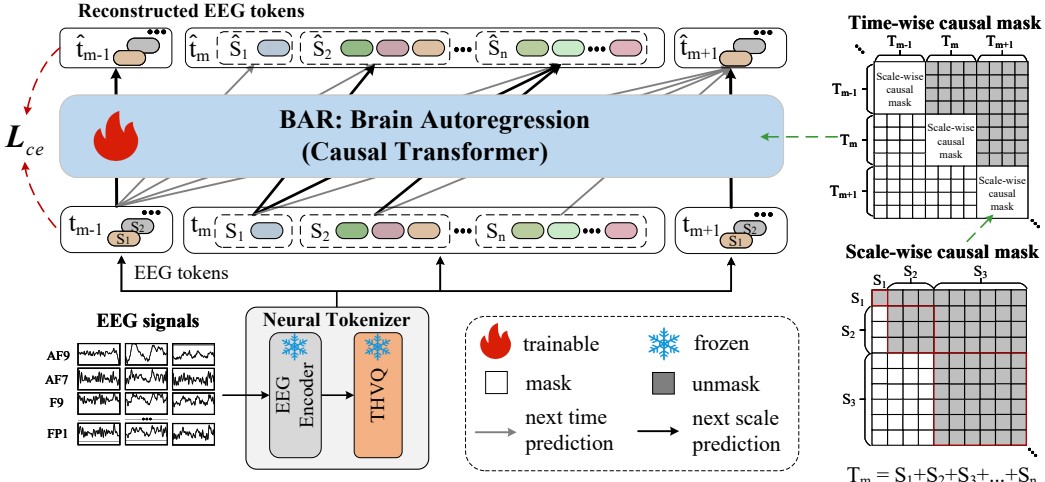

Figure 4: A BAR causal transformer is trained via "next-scale-time prediction".

---

**Algorithm 1** : THVQ-VAE Encoding and Quantization

1: **Inputs:** EEG signal $eeg \in \mathbb{R}^{N \times T \times P}$;
2: **Hyperparameters:** scales $S$, multi-scale channel hierarchy $(ch_s)_{s=1}^S$;
3: $f = \mathcal{E}(eeg) \in \mathbb{R}^{N \times T \times C}, R = []$;
4: $f = reshape(f, (N * T, C))$;
5: **for** $s = 1, \ldots, S$ **do**
6: $\quad r_s = \mathcal{Q}(downscale(f, ch_S, ch_s))$;
7: $\quad R = queue\_push(R, r_s)$;
8: $\quad z_s = lookup(Z, r_s)$;
9: $\quad z_s = upscale(z_s, ch_s, ch_S)$;
10: $\quad f = f - \phi_s(z_s)$;
11: **end for**;
12: **return** multi-scale tokens $R$;

---

**Algorithm 2** : THVQ-VAE Dequantization and Decoding

1: **Inputs:** multi-scale token maps $R$;
2: **Hyperparameters:** scales $S$, multi-scale channel hierarchy $(ch_s)_{s=1}^S$;
3: $\hat{f} = 0$;
4: **for** $s = 1, \ldots, S$ **do**
5: $\quad r_s = queue\_pop(R)$;
6: $\quad z_s = \mathcal{DQ}(Z, r_s)$;
7: $\quad z_s = upscale(z_s, ch_s, ch_S)$;
8: $\quad \hat{f} = \hat{f} + \phi_s(z_s)$;
9: **end for**
10: $\hat{f} = reshape(f, (N, T, C))$;
11: $e\hat{e}g = \mathcal{D}(\hat{f})$;
12: **return** reconstructed EEG signals $e\hat{e}g$;

---

## 2.3 Brain Autoregressive Pre-training

**Preliminary: Autoregressive Modeling via Next-Time Prediction.** Consider discrete token maps $q^t = \{q^{(n,t)} | n = 1, 2, ..., N\}$ of the VQ-VAE, where each $q^{(n,t)} \in [V]$ is an integer from a vocabulary of size $V$. The "next-time" autoregressive model assumes that the probability of observing the current token $q^t$ depends on its prefix $(q^1, q^2, ..., q^{t-1})$. This unidirectional token dependency enables the factorization of the sequence likelihood of $q$:

$$P(q^1, q^2, \ldots, q^T) = \prod_{t=1}^{T} P(q^t \mid q^1, q^2, \ldots, q^{t-1}) \tag{4}$$

The "next-time prediction" means training the autoregressive model $P_\theta$ through optimizing $P(q^t | q^1, q^2, ..., q^{t-1})$ over datasets. During the implementation, we define time-wise mask that allows each EEG token to attend to tokens of all channels at the previous $1 \sim (t-1)$ times.

**Autoregressive Modeling via Next-Scale-Time Prediction.** Consider multi-scale discrete token maps $q^t = \{r_s^t | s = 1, 2, ..., S\}$, where each element of $r_s^t$ is an integer from a vocabulary of size $V$. Our BAR module, as shown in Figure 4, employs a "next-scale-time prediction" strategy and assumes that the probability of observing the current token $r_s^t$ depends on its prefix $(q^1, q^2, ..., q^{t-1})$ and $(r_1^t, r_2^t, ..., r_{s-1}^t)$. This causal token dependency enables the factorization of the sequence likelihood of $q$:

$$P(q^1, q^2, ..., q^T) = \prod_{t=1}^{T} \prod_{s=1}^{S} P(r_s^t | \underbrace{q^1, q^2, ..., q^{t-1}}_{1 \sim (t-1) \text{ times}}, \underbrace{r_1^t, r_2^t, ..., r_{s-1}^t}_{1 \sim (s-1) \text{ scales}}) \tag{5}$$

The "next-scale-time prediction" means training the autoregressive model $P_\theta$ through optimizing $P(r_s^t | q^1, q^2, ..., q^{t-1}, r_1^t, r_2^t, ..., r_{s-1}^t)$ over datasets. During the implementation, we define scale-time-wise mask that allows each EEG token to attend to tokens of all scales at the previous $1 \sim (t-1)$ times and tokens of $1 \sim (s-1)$ scales at current $t$ time.

## 2.4 Multi-task Instruction Fine-tuning

We employ joint multi-task instruction fine-tuning to handle various EEG downstream datasets, adopting a fine-tuning strategy similar to that used in [6]. The specific design of instructions for each dataset is detailed in Appendix F. EEG and text data are concatenated using a special token, [SEP], to distinguish between the two modalities. The loss is computed based on the answer portion of the text, which corresponds to the classification result. Let $prom$ represent the instruction prompt and $t^a$ the answer to the instruction. Let $L$ denote the sequence length of $t^a$. This procedure can be expressed as follows:

$$p(t^a|prom) = \prod_{i=1}^{L} p(t_i^a|prom, t_{,<i}^a),$$
(6)

where $t_{,<i}^a$ represents the answer tokens that occur before the current prediction token $t_i^a$.

## 3 Datasets and Implementation Details

### 3.1 Datasets

The THD-BAR framework is evaluated through a two-stage process: pre-training and subsequent fine-tuning on downstream tasks. For pre-training our THVQ-VAE tokenizer and BAR module, we utilize a comprehensive corpus of 17 diverse EEG datasets, which are detailed in Appendix B. To comprehensively assess the generalization capabilities of the pre-trained THD-BAR framework, we then employ 10 distinct EEG datasets for fine-tuning and evaluation across 5 major downstream tasks, summarized in Table 1. These five downstream task categories are: emotion recognition (DEAP[17], SEED[18]), motor imagery recognition (MIBCI[19], BCIC4-1[20]), mental workload recognition (EEGMat[21], STEW[22]), sleeping stage recognition (EDF[23], HMC[24]), and epilepsy recognition (TUAB[25], TUEV[25]). Further descriptions and processing details for these downstream datasets are available in Appendix C.

Table 1: Summary of EEG datasets used in downstream tasks.

| Task | Dataset | Rate | Subject | Electrode | Sample | Time | Class |
|---|---|---|---|---|---|---|---|
| Emotion | DEAP | 128Hz | 32 | 32 | 19.2k | 60s | 4 |
| Recognition | SEED | 1000Hz | 15 | 62 | 36.2k | 4s | 3 |
| Motor Imagery | MIBCI | 512Hz | 52 | 64 | 10.5k | 7s | 2 |
| Recognition | BCIC4-1 | 100Hz | 7 | 38 | 1.4k | 8s | 2 |
| Mental Workload | EEGMat | 500Hz | 34 | 19 | 1.0k | 60s | 2 |
| Recognition | STEW | 128Hz | 45 | 14 | 3.3k | 4s | 3 |
| Sleeping Stage | EDF | 100Hz | 78 | 2 | 19.5k | - | 5 |
| Recognition | HMC | 256Hz | 151 | 4 | 22.6k | 30s | 5 |
| Epilepsy | TUAB | 256Hz | 2383 | 23 | 409.5k | 10s | 2 |
| Recognition | TUEV | 256Hz | 370 | 23 | 112.5k | 5s | 6 |

### 3.2 Implementation Details

**Data Preprocessing.** Due to variations in data collection equipment, sampling parameters, and noise interference, we used essential preprocessing steps. A band-pass filter with cutoff frequencies of 0.1 Hz and 75 Hz is applied to remove low and high-frequency noise, while a 50/60 Hz notch filter is employed to eliminate power-line interference. All EEG signals are sampled to 200 Hz, and the interquartile range (IQR) is used for robust scaling to reduce the outlier influence and ensure stable data normalization.

**Model Configurations.** The implementation of the encoder and decoder in THVQ-VAE adopts the vanilla Transformer [26] following LaBraM [5] and NeuroLM [6]. We predict both EEG signals and their frequency magnitude using two identical decoders. BAR module adopts GPT-2 series as large language model, which is compatible with any causal LLM. We have developed three architecture configurations: THD-BAR-Base, THD-BAR-Large, THD-BAR-Huge, which have 124M, 354M, 1555M parameters, respectively. The patch size $P$ and window size $W$ are set to 200 and 1024. Sequences shorter than 1024 are padded with zeros to reach this length during tokenizer training and autoregressive pre-training, with attention values for the padding masked.

Table 2: Comparative performance (balanced accuracy %) and model parameters on emotion recognition and motor imagery tasks. "General Model?" indicates if the model is pre-trained for general representations, and "Multi-Task?" indicates if it's designed for or evaluated on multiple tasks. Best results are in **bold**.

| Methods | Year | General Model? | Multi-Task? | Model Parameter | Emotion | | Motor Imagery | |
|---|---|---|---|---|---|---|---|---|
| | | | | | DEAP | SEED | MIBCI | BCIC4-1 |
| EEGNet [27] | 2018 | ✗ | ✗ | - | 35.2±9.4 | 69.3±2.1 | 63.3±7.2 | 51.9±1.5 |
| TSception [28] | 2020 | ✗ | ✗ | - | 34.3±8.1 | 68.6±1.3 | 61.4±6.5 | 52.2±1.6 |
| LGGNet [29] | 2024 | ✗ | ✗ | - | 33.5±8.5 | 69.5±1.4 | 56.7±3.7 | 50.0±0.4 |
| BIOT [7] | 2023 | ✓ | ✗ | 3.2M | 35.2±8.9 | 71.0±0.2 | 53.2±2.0 | 51.1±0.5 |
| LaBraM [5] | 2024 | ✓ | ✗ | 5.8M | 34.3±9.9 | 73.2±0.2 | 50.5±1.1 | 50.3±0.4 |
| EEGPT [8] | 2024 | ✓ | ✓ | 1.46M | 41.4±2.7 | - | 62.2±2.8 | 56.9±1.6 |
| NeuroLM [6] | 2024 | ✓ | ✓ | 254M | 40.1±1.4 | 70.2±0.3 | 62.1±2.6 | 57.1±1.8 |
| THD-BAR-Base | 2025 | ✓ | ✓ | 124M | 42.3±1.2 | 73.5±0.3 | 62.9±1.4 | 57.5±0.9 |
| THD-BAR-Large | 2025 | ✓ | ✓ | 354M | 43.6±1.6 | 73.4±0.4 | 63.3±1.7 | 58.1±1.2 |
| THD-BAR-Huge | 2025 | ✓ | ✓ | 1555M | **43.9**±1.7 | **73.9**±0.3 | **63.6**±1.8 | **58.9**±1.5 |

Table 3: Comparative Performance (balanced accuracy %) on Mental Workload, Sleep Staging, and Epilepsy Detection Tasks. Best results are in **bold**.

| Methods | General Model? | Multi-Task? | Mental Workload | | Sleeping Stage | | Epilepsy | |
|---|---|---|---|---|---|---|---|---|
| | | | EEGMat | STEW | EDF | HMC | TUAB | TUEV |
| EEGNet [27] | ✗ | ✗ | 60.0±8.7 | 52.3±17.6 | 84.0±4.4 | 54.5±8.7 | 76.3±1.5 | 53.5±0.2 |
| TSception [28] | ✗ | ✗ | 50.3±1.2 | **63.8**±13.0 | 68.6±4.5 | 36.4±9.8 | 74.3±4.2 | 51.3±0.4 |
| LGGNet [29] | ✗ | ✗ | 50.2±1.1 | 46.7±12.5 | 68.6±4.5 | 17.0±9.5 | 75.5±3.1 | 52.8±0.3 |
| BIOT [7] | ✓ | ✗ | 50.2±1.1 | 51.3±11.9 | 69.4±4.6 | 63.0±1.1 | 79.6±0.6 | 52.8±0.3 |
| LaBraM [5] | ✓ | ✗ | 50.4±1.3 | 52.5±12.4 | 69.3±3.8 | 68.1±0.7 | 81.4±0.2 | 64.1±0.7 |
| EEGPT [8] | ✓ | ✓ | 66.0±8.6 | 63.2±10.6 | 85.2±3.4 | 65.5±4.0 | - | - |
| NeuroLM [6] | ✓ | ✓ | 65.7±7.5 | 59.3±5.8 | 85.3±3.7 | 67.4±5.6 | 78.3±0.5 | 45.6±0.6 |
| THD-BAR-Base | ✓ | ✓ | 66.5±6.2 | 62.1±6.7 | 85.5±4.6 | 68.0±3.5 | 81.9±0.4 | 64.3±0.2 |
| THD-BAR-Large | ✓ | ✓ | 66.4±7.1 | 62.4±7.5 | **85.7**±4.7 | 68.0±3.2 | 82.0±0.3 | 64.9±0.3 |
| THD-BAR-Huge | ✓ | ✓ | **67.1**±5.8 | 62.9±7.9 | **85.7**±4.4 | **68.4**±4.3 | **82.2**±0.4 | **65.3**±0.5 |

**Training Settings.** All experiments are conducted using Python 3.12.9, Pytorch 2.5.0, and CUDA 12.2 on a system equipped with 8 NVIDIA L40s-48G GPUs. Further experimental configuration details are available in Appendix E.

# 4 Experimental Results

## 4.1 Comparative Study

We conducted comprehensive evaluation experiments for our THD-BAR framework on 10 downstream EEG datasets, encompassing 5 diverse tasks. As shown in Table 2 and Table 3, our proposed THD-BAR models consistently demonstrate superior performance, outperforming both multi-task and single-task baselines across 9 of the 10 evaluated downstream EEG datasets. This highlights THD-BAR as a highly competitive method, often surpassing state-of-the-art approaches across a variety of EEG tasks. Notably, compared to the multi-task baseline NeuroLM [6], our THD-BAR-Base model achieves improved balanced accuracy across all evaluated datasets, especially achieving significant balanced accuracy improvements of 2.2% on DEAP and 3.3% on SEED, highlighting the advancements THD-BAR brings to existing AR capabilities. Our models perform slightly worse than EEGPT [8] on the STEW dataset. This can be attributed to EEGPT benefiting from both pre-training and fine-tuning on the STEW dataset, whereas our approach was only fine-tuned on STEW after general pre-training. Overall, the results demonstrate that THD-BAR is a highly competitive method. Furthermore, our experiments consistently show that increasing the number of parameters within the THD-BAR architecture (from Base to Large to Huge) generally leads to improved performance.

## 4.2 Ablation Study

**Multi-scale ablation.** To validate the necessity of our multi-scale design and identify optimal BTH scale utilization, we conducted an ablation study examining the impact of various scale combinations. We evaluated the performance of the THVQ-VAE tokenizer and the BAR model across 9 distinct scale configurations ($O_1$ to $O_9$). Figure 5 displays the BTH scales ($S_1$ to $S_5$)

for each configuration ($O_1$ to $O_9$) (center), and the corresponding THVQ-VAE (PCC, Loss; top) and BAR module (ACC, Loss; bottom) performance, where configuration $O_5$ is optimal for both. These results consistently indicate that configuration $O_5$, which incorporates all scales from $S_1$ to $S_5$, yields the optimal performance for both the THVQ-VAE and BAR under the evaluated metrics. Comparing configurations using only single scales (e.g., $O_1$ using only the coarsest scale $S_1$, or $O_9$ using only the finest scale $S_5$) resulted in significantly lower performance than configurations using multiple scales, suggesting that relying on a single granularity level fails to capture the full complexity of EEG signals that a multi-scale representation can provide. This highlights that the integration of information across the full spectrum of defined scales, as achieved in configuration $O_5$, is crucial for achieving optimal performance in both the tokenization and autoregressive modeling stages.

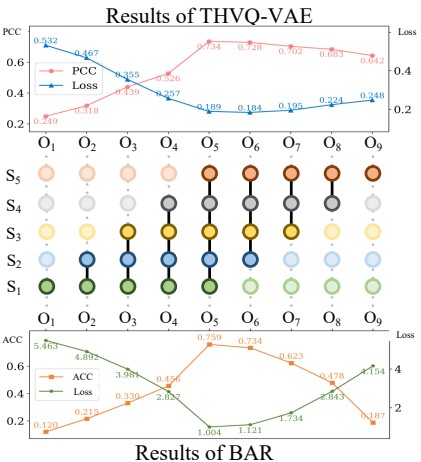

Figure 5: Performance comparison of THVQ-VAE and BAR across different multi-scale configurations.

**Mask design ablation.** To determine the optimal masking strategy for our "next-scale-time prediction" within the THD-BAR framework, we examined three mask variants, as illustrated in Figure 6 (a-c): **Scale-wise Mask** conceals all tokens from other time steps and finer-scale tokens within the current time step, compelling prediction based only on revealed coarser-scale tokens within that same time step. **Time-wise Mask** enforces strict temporal causality by concealing tokens from all future time steps, prompting the model to primarily learn from tokens in previous time steps. **Scale-Time-wise Mask** implements our full "next-scale-time prediction" strategy by combining scale-wise spatial prediction within the current time step and masking all future time steps. The results, presented in Figure 6 (d-f), demonstrate that the mask design significantly impacts autoregressive modeling, with the Scale-Time-wise mask yielding the best overall performance. This suggests that our proposed nested "next-scale-time prediction" strategy is effective for learning the complex spatio-temporal dependencies inherent in EEG data.

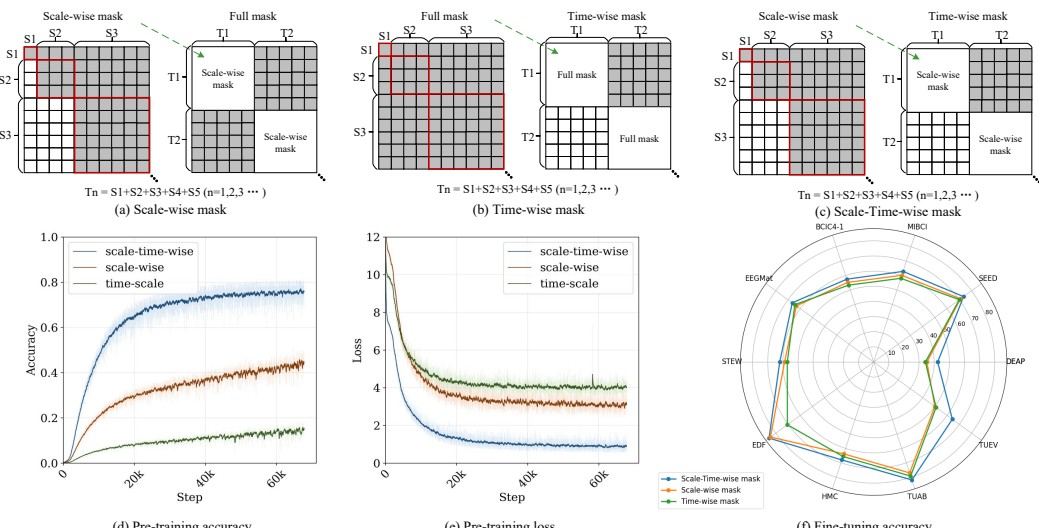

Figure 6: Mask design ablation study. (a-c) Illustrations of Scale-wise, Time-wise, and Scale-Time-wise masking strategies. (d-e) Pretraining accuracy and loss curves for the three mask variants. (f) Fine-tuning accuracy comparison across various downstream tasks. The Scale-Time-wise mask consistently yields the best performance.

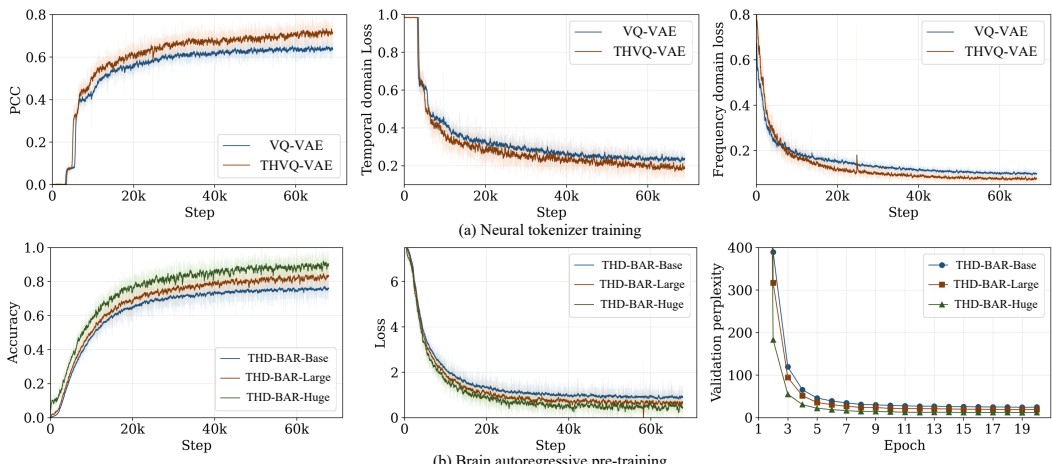

Figure 7: Training performance comparison. (a) Neural tokenizer training dynamics comparing THVQ-VAE and VQ-VAE on PCC, temporal domain loss, and frequency domain loss. (b) Brain autoregressive pre-training performance comparing THD-BAR-Base, THD-BAR-Large, and THD-BAR-Huge models on accuracy, training loss, and validation perplexity.

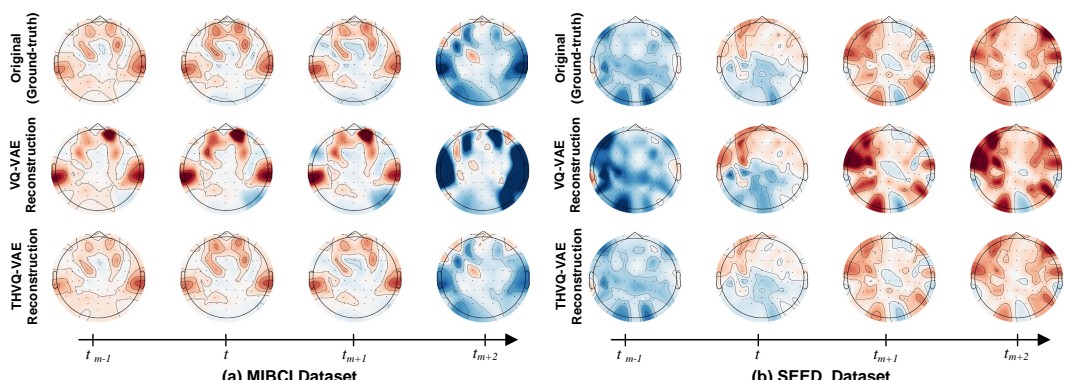

Figure 8: Qualitative comparison of spatio-temporal EEG reconstruction quality. Spatial heatmaps depict original EEG (Ground-truth), VQ-VAE reconstructed EEG, and THVQ-VAE reconstructed EEG across four consecutive time steps ($t_{m-1}$, $t_m$, $t_{m+1}$, $t_{m+2}$).

## 4.3 Visualization Study

In Figure 7 (a), we visualize the PCC, temporal domain loss, and frequency domain loss during neural tokenizer training. Our comparison between THVQ-VAE and VQ-VAE used in NeuroLM [6] shows that the PCC improved by over 8%. Additionally, Figure 7 (b) illustrates the accuracy, loss, and validation perplexity of brain autoregressive pre-training. Generally, larger models tend to achieve lower loss and perplexity.

To validate the spatio-temporal feature extraction capabilities of our framework, Figure 8 visualizes THVQ-VAE's ability to capture the dynamic evolution of brain activity patterns across consecutive time steps ($t_{m-1}$ to $t_{m+2}$) for representative samples from the MIBCI and SEED datasets. THVQ-VAE's reconstructions more faithfully mirror the temporal progression and transformation of spatial topographies compared to the baseline VQ-VAE. This superior correspondence in evolving patterns suggests our method extracts more comprehensive spatio-temporal features, effectively representing the underlying dynamic characteristics of EEG signals. More visualization results can be found in Appendix G.

# 5 Conclusion

This paper proposes THD-BAR, a novel autoregressive framework for EEG signals based on "next-scale-time prediction". We first established a BTH, rooted in physiological principles. Subsequently, a THVQ-VAE encodes EEG signals into BTH-aligned multi-scale discrete tokens. THD-BAR leverages a "scale-time-wise mask" to facilitate spatio-temporal prediction. After pre-training on 17 datasets and validating on 10 downstream datasets across 5 tasks, our proposed method demonstrated superior performance and generalization. Unlike conventional "next-token prediction" approaches, THD-BAR significantly improves spatial feature modeling while maintaining robust temporal dynamics. This hierarchical framework presents new perspectives for BCI research, and the open-source code is intended to support further EEG foundation model development.

## Acknowledgments

This work was supported in part by the National Natural Science Foundation of China under Grant 62325301, Grant 623B2011, and Grant U24B20186; in part by the Beijing Natural Science Foundation under Grant Z220017; in part by the National Key Research and Development Program of China under Grant 2023YFC2416600; in part by Natural Science Foundation Key Project of Zhejiang Province, China under Grant LZ23F030001; and in part by the Chongqing Municipal Health Commission, China under Grant 2025GGXM005. It was also supported by the Academic Excellence Foundation of BUAA for PhD students.

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

# A   Related Work

The pursuit of generalizable representations from Electroencephalography (EEG) has spurred significant research into large-scale pre-training, drawing inspiration from successes in other domains. This section elaborates on the evolution of these efforts and the specific challenges that persist, particularly concerning the modeling of EEG's inherent spatio-temporal complexity. Early large-scale EEG models primarily focused on mitigating superficial data inconsistencies. For instance, MMM[4] pioneered techniques to handle variable electrode configurations by employing a masked autoencoder with region-wise tokenization, aiming to learn robust spatial features irrespective of specific montage details. This work highlighted the importance of addressing spatial heterogeneity directly in the pre-training phase. Subsequent research broadened the scope, incorporating architectural innovations and more sophisticated pre-training objectives. BIOT[7], for example, introduced a versatile tokenization scheme to adapt biosignals of varying lengths and types into a unified "sentence-like" structure, facilitating cross-task learning. LaBraM[5] advanced the application of Masked Autoencoders (MAE) by focusing on spectral prediction in the quantized space, tokenizing raw EEG through vector-quantized representations of their frequency content. This line of work underscored the potential of self-supervision in learning meaningful EEG features without explicit labels. The development of autoregressive (AR) models marked another significant step. NeuroLM[6] scaled up model capacity substantially by adopting a GPT-2[2] series architecture and introduced multi-modal alignment with text, enhancing the model's ability to generalize across a wider array of EEG tasks. Concurrently, EEGPT[8] explored dual-alignment strategies within self-supervised frameworks, combining spatio-temporal representation alignment with mask-based signal reconstruction to improve feature quality and training stability. These AR-based approaches demonstrated impressive capabilities in learning from large unlabeled EEG corpora. Despite these advancements in handling input variability, integrating modalities, and scaling model size, a fundamental aspect often remains under-explored in existing AR frameworks: the optimal sequential representation of multi-channel EEG that respects its underlying physiological organization. While powerful, the predominant "next-time prediction" paradigm, where multi-channel data is typically processed as a sequence of temporal snapshots, does not inherently account for the brain's hierarchical topological structure or the dynamic, multi-scale interactions across different brain regions. Capturing these intricate spatio-temporal dependencies—how activity at different spatial granularities co-evolves and influences future states across both scales and time—presents an ongoing challenge. This nuanced requirement for a more physiologically-grounded sequential ordering forms the primary motivation for exploring novel hierarchical autoregressive approaches, such as the one proposed in this work.

# B   Pre-training Dataset Description

The comprehensive description of the 17 datasets utilized for pre-training in our study is as follows:

- **SEED-IV** [30]: This dataset includes 15 subjects, with EEG signals recorded using the ESI NeuroScan system with 62 channels at a sampling rate of 1000 Hz. SEED-IV included four categories: happy, sad, fear, and neutral, totaling 8.43 hours.

- **SEED-V** [31]: Including 15 subjects, this dataset was collected using the same setup as SEED-V (62 channels, 1000 Hz, ESI NeuroScan system). SEED-V expanded categories to five distinct emotions: happy, sad, fear, disgust and neutral, totaling 41.55 hours.

- **SEED-GER** [32]: This dataset contains EEG recordings from 8 German subjects who viewed emotionally charged video stimuli. EEG data were collected using the ESI NeuroScan system, with emotions classified as positive, negative, and neutral, totaling 25.92 hours.

- **SEED-FRA** [32]: EEG signals from 8 French participants were recorded using the same methodology as SEED-GER. While watching emotion-related videos, subjects' EEG data were captured via the ESI NeuroScan system, with emotions categorized into positive, negative, and neutral, totaling 25.48 hours.

- **EmoBrain** [33]: This multimodal emotion dataset consists of EEG recordings from 16 participants, captured with 64 channels at 1024 Hz using the Biosemi Active 2 system. Emotional stimuli were derived from a selected subset of the IAPS dataset, totaling 4.94 hours.

- **Grasp and Lift EEG Challenge** [34]: The dataset contains EEG signals (32 channels, 500 Hz) from 12 subjects engaged in grasp-and-lift (GAL) trials. EEG recordings were acquired using a BrainAmp EEG signal amplifier, totaling 11.72 hours.

- **Inria BCI Challenge** [19]: A dataset focusing on P300-based spelling, featuring EEG signals (56 channels, 600 Hz) collected from 26 individuals. EEG data were recorded using Ag/AgCl EEG sensors with a VSM-CTF compatible system, totaling 29.98 hours.

- **EEG Motor Movement/Imagery Dataset** [35]: This dataset includes motor imagery EEG recordings from 109 volunteers using 64 channels at 160 Hz. The experiment involved baseline tasks (eyes open/closed), motor movements, and imagery tasks (both fists or both feet), recorded with the BCI2000 system, totaling 47.3 hours.

- **Raw EEG Data** [36]: EEG signals from a categorization task, recorded at 64 channels with a 256 Hz sampling rate. The dataset includes data from an Information-Integration categorization task and a multidimensional Rule-Based categorization task, totaling 34.35 hours.

- **Resting State EEG Data** [37]: EEG data from 22 participants, who engaged in an 8-minute resting task (4 minutes with eyes closed, 4 minutes with eyes open). The data were recorded with 64 EEG channels at 256 Hz, using BioSemi caps or freestanding Ag/AgCl electrodes, totaling 3.04 hours.

- **Siena Scalp EEG Database** [38]: EEG recordings from 14 patients, collected using EB Neuro and Natus Quantum LTM amplifiers with reusable silver/gold cup electrodes. The dataset consists of 31-channel recordings at 512 Hz, totaling 30.47 hours.

- **SPIS Resting State Dataset** [39]: This dataset features EEG recordings from 10 individuals, captured with 64 channels at 2048 Hz. Participants completed 2.5-minute eyes-closed and eyes-open sessions before engaging in a 105-minute Sustained Attention to Response Task (fixed-sequence with varying ISIs), totaling 0.83 hours.

- **Target Versus Non-Target** [40]: EEG recordings from 50 participants playing Brain Invaders, a visual P300 brain-computer interface game employing an oddball paradigm with adaptive Riemannian geometry and no calibration requirement. EEG signals were collected from 32 channels at 512 Hz using a g.USBamp amplifier and g.GAMMAcap, totaling 16 hours.

- **TUAR** [41]: A dataset consisting of EEG recordings annotated with five types of artifacts, recorded with 23 channels at 256 Hz, totaling 92.22 hours.

- **TUEP** [42]: This dataset contains EEG recordings from 200 participants—100 diagnosed with epilepsy and 100 without. Data were verified by a certified neurologist, with recordings captured using 19-23 channels at 256 Hz, totaling 591.22 hours.

- **TUSZ** [43]: A manually annotated EEG dataset for seizure detection, including precise start and stop times, affected channels, and seizure classifications. EEG recordings were obtained using 19-23 channels at 256 Hz, totaling 1138.53 hours.

- **TUSL** [44]: EEG recordings annotated for slowing events, using 23 channels at 256 Hz. This dataset has been employed in research on common errors in automated seizure detection, totaling 20.59 hours.

## C   Multi-task Dataset Description

The comprehensive description of the 10 datasets utilized for multi-task instruction fine-tuning in our study is as follows:

- **DEAP** [17]: This dataset is an emotion recognition dataset with 32-channel EEG recordings from 32 participants at 128 Hz. Emotions are classified into four categories based on high/low valence and arousal.

- **SEED** [18]: This dataset comprises EEG recordings from 15 participants, captured using the ESI NeuroScan system with 62 channels at a 1000 Hz sampling rate. Participants watched emotion-related videos to induce three emotional states: positive, negative, and neutral.

- **MIBCI** [19]: MIBCI is a motor imagery EEG dataset with 64-channel recordings from 52 participants, sampled at 512 Hz. It supports binary classification based on motor imagery of the left and right hands.

- **BCIC4-1** [20]: A motor imagery EEG dataset featuring recordings from seven individuals. EEG data were collected via BrainAmp MR plus amplifiers and Ag/AgCl electrode caps, with 59 EEG channels at 1000 Hz sampling rate. The study included motor imagery tasks for the left hand, right hand, foot, and an idle state, totaling 8.21 hours.

- **EEGMat** [21]: EEGMat is a mental workload dataset featuring 23-channel EEG recordings from 36 participants, sampled at 500 Hz. It includes two states: rest and task performance, supporting a binary classification for mental workload detection.

- **STEW** [22]: STEW includes 14-channel EEG data recorded at 128 Hz from 45 participants. It covers three levels of mental workload: low, medium, and high. This allows for a three-class classification task to detect mental workload based on EEG signals.

- **EDF** [23]: This dataset is a sleep stage classification dataset with 2-channel EEG recordings from 78 participants, sampled at 100 Hz. It includes five sleep stages: wake, N1, N2, N3, and movement, supporting a five-class classification task.

- **HMC** [24]: This dataset is a 4-channel EEG dataset for automatic sleep stage classification, recorded at 256 Hz from 151 participants. It covers five sleep stages: wake, N1, N2, N3, and REM, supporting a five-class classification task.

- **TUAB** [25]: This dataset is a 32-channel EEG dataset for epilepsy abnormality detection, sampled at 256 Hz. It consists of 10-second segments and supports binary classification of clinically normal and abnormal signals.
- **TUEV** [25]: This dataset is EEG dataset for event type classification, sampled at 256 Hz. It consists of 5-second segments and supports six-class classification, covering periodic lateralized epileptiform discharge, generalized periodic epileptiform discharge, spike/sharp wave discharges, artifacts, eye movement, and background.

## D  Brain Topology Hierarchy

This appendix provides further details on the Brain Topology Hierarchy (BTH) introduced in Section 2 (specifically, subsection 2.1), with a focus on the exemplary 5-scale configuration illustrated in Figure 9 and utilized throughout our experiments. The BTH is designed to provide a structured, multi-scale spatial ordering for EEG channels, moving from global brain representations to individual channel details.

The 5-scale BTH depicted in Figure 9 is constructed as follows:

- **Scale S1 (Whole Brain):** This represents the coarsest level of the hierarchy. At S1, all EEG channels are considered as a single unit, providing a global representation of overall brain activity. This scale allows the model to capture widespread, synchronous neural events or global brain state features.
- **Scale S2 (Major Brain Regions):** At S2, the brain is parcellated into a few broad regions. These regions are typically defined based on the spatial proximity of channels and often align with general anatomical divisions. For example, common groupings might approximate anterior (frontal), central (parietal/sensorimotor), and posterior (occipital/temporal-parietal) areas. This level enables the modeling of large-scale inter-regional interactions and broader functional specializations. In our specific implementation shown, S2 divides the channels into three primary horizontal bands reflecting these coarse regional distinctions.
- **Scale S3 (Sub-regions):** Scale S3 further refines the broad regions defined in S2 into smaller, more localized zones. Each major region from S2 is subdivided, allowing for a more granular analysis of brain activity. For instance, an anterior region might be split into left-frontal, mid-frontal, and right-frontal sub-regions. This level helps in capturing more specific regional activations and their interplay. Figure 9 illustrates this further parcellation.
- **Scale S4 (Channel Clusters):** This scale continues the hierarchical decomposition, providing even finer spatial granularity. The sub-regions from S3 are divided into smaller clusters of channels. These clusters might correspond to more specific functional parcels or highly localized groups of electrodes, enabling the model to learn features reflecting very localized neural processing.
- **Scale S5 (Individual Channels):** Scale S5 represents the finest level of the hierarchy used in our model. At this scale, each node or unit effectively corresponds to an individual EEG channel, or a very small, highly localized group if the original channel density is extremely high. This allows for the most detailed spatial resolution, capturing channel-specific information and fine-grained spatial patterns.

The rationale behind this 5-scale structure is to provide a rich, hierarchical representation that allows the THD-BAR framework to learn and integrate features across multiple spatial resolutions simultaneously. This progressive refinement from global (S1) to local (S5) information is hypothesized to be crucial for capturing the complex and multi-faceted nature of brain dynamics, where different cognitive processes might manifest at different spatial scales.

The specific groupings of channels at each intermediate scale (S2-S4) are primarily guided by the spatial adjacency of channels based on standard EEG montages (e.g., extensions of the 10-20 system) and aim to reflect plausible functional or anatomical parcellations where possible. While the BTH concept is flexible and the number of scales or the exact channel groupings can be adapted based on specific EEG hardware, channel density, or research questions, the 5-scale hierarchy presented here provides a robust and physiologically-inspired framework for our experiments. This structured approach to spatial ordering is a key component enabling the "next-scale-time prediction" strategy of our THD-BAR model.

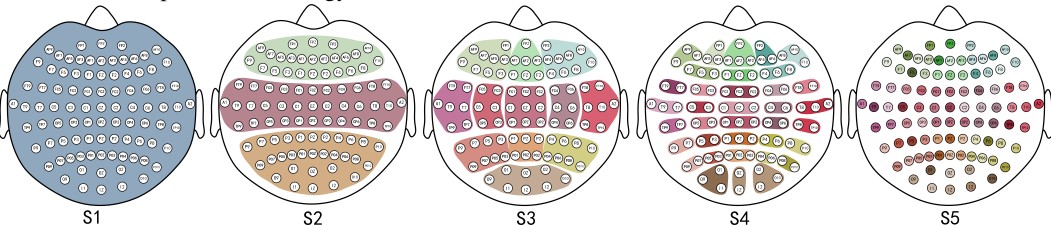

Figure 9: Brain Topology Hierarchy (BTH).

# E Experimental Configurations

Table 4: Hyperparameters for neural tokenizer.

| Hyperparameters | Values |
|---|---|
| **Temporal Encoder** | |
| Input channels | {1, 16, 16} |
| Output channels | {16, 16, 16} |
| Kernel size | {15, 3, 3} |
| Stride | {8, 1, 1} |
| Padding | {7, 1, 1} |
| Transformer encoder layers | 12 |
| Transformer decoder layers | 3 |
| Hidden size | 768 |
| MLP size | 3072 |
| Attention head number | 12 |
| Codebook size | $8192 \times 128$ |
| **Training** | |
| Batch size | 512 |
| Peak learning rate | 5e-5 |
| Minimal learning rate | 1e-5 |
| Learning rate scheduler | Cosine |
| Optimizer | AdamW |
| Adam $\beta$ | (0.9, 0.999) |
| Weight decay | 1e-4 |
| Total epochs | 50 |
| Warmup epochs | 5 |
| Data overlap | None |
| Gradient clipping | None |

Table 5: Hyperparameters for autoregressive pre-training.

| Hyperparameters | THD-BAR-Base | THD-BAR-Large | THD-BAR-Huge |
|---|---|---|---|
| Model size | 124M | 354M | 1555M |
| Transformer encoder layers | 12 | 24 | 48 |
| Hidden size | 768 | 1024 | 1600 |
| MLP size | 3072 | 4096 | 6400 |
| Attention head number | 12 | 16 | 25 |
| EEG batch size | 480 | 512 | 512 |
| Text batch size | 32 | 64 | 64 |
| Peak learning rate | 6e-4 | 6e-4 | 6e-4 |
| Minimal learning rate | 6e-5 | 6e-5 | 6e-5 |
| Learning rate scheduler | Cosine | Cosine | Cosine |
| Optimizer | AdamW | AdamW | AdamW |
| Adam $\beta$ | (0.9, 0.95) | (0.9, 0.95) | (0.9, 0.95) |
| Weight decay | 0.1 | 0.1 | 0.1 |
| Total epochs | 20 | 20 | 20 |
| Warmup epochs | 2 | 2 | 2 |
| Data overlap | None | None | None |
| Gradient clipping | 1 | 1 | 1 |

Table 6: Hyperparameters for instruction tuning.

| Hyperparameters | Values |
|---|---|
| Instruction batch size | 512 |
| Text batch size | 128 |
| Peak learning rate | 5e-4 (B), 5e-5 (L), 2e-5 H) |
| Minimal learning rate | 5e-5 (B), 5e-6 (L), 2e-6 (H) |
| Learning rate scheduler | Cosine |
| Optimizer | AdamW |
| Adam $\beta$ | (0.9, 0.95) |
| Weight decay | 0.1 |
| Total epochs | 5 (B, L), 3 (H) |
| Warmup ratio | 0.1 |
| Gradient clipping | 1 |

# F  Instruction Description

Table 7: Instruction description for downstream datasets.

| Dataset | Instruction Description |
|---|---|
| **DEAP** | [SEP] Question: What are the valence and arousal levels of EEG segment? Options: (A) Low valence and low avoidance. (B) High valence and low avoidance. (C) Low valence and high avoidance. (D) High valence and high avoidance. Answer: {(A), (B), (C), (D)} [END] |
| **SEED** | [SEP] Question: Which emotion type does this EEG segment belong to? Options: (A) Positive, (B) Neutral, (C) Negative. Answer: {(A), (B), (C)} [END] |
| **MIBCI** | [SEP] Question: Is this EEG segments for Left hand or right hand motor imagery? Options: (A) Left, (B) Right. Answer: {(A), (B)} [END] |
| **BCIC4-1** | [SEP] Question: Is this EEG segments for Left hand or right hand motor imagery? Options: (A) Left, (B) Right. Answer: {(A), (B)} [END] |
| **EEGMat** | [SEP] Question: Is this EEG segment for rest or for task? Options: (A) Rest, (B) Task. Answer: {(A), (B)} [END] |
| **STEW** | [SEP] Question: What is the mental workload level of this EEG segment? Options: (A) Low, (B) Medium, (C) High. Answer: {(A), (B), (C)} [END] |
| **EDF** | [SEP] Question: What is the sleep stage of this EEG segment? Options: (A) wake, (B) N1. (C) N2. (D) N3. (E) Movement. Answer: {(A), (B), (C), (D), (E)} [END] |
| **HMC** | [SEP] Question: Which sleep type does this EEG segment belong to? Options: (A) Wake. (B) NREM-1. (C) NREM-2. (D) NREM-3. (E) REM. Answer: {(A), (B), (C), (D), (E)} [END] |
| **TUAB** | [SEP] Question: Is this EEG segment abnormal? Answer: {Yes, No} [END] |
| **TUEV** | [SEP] Question: Which event type does this EEG segment belong to? Options: (A) spike and slow wave. (B) generalized periodic epileptiform discharge. (C) periodic lateralized epileptiform discharge. (D) eye movement. (E) artifact. (F) background. Answer: {(A), (B), (C), (D), (E), (F)} [END] |

# G  Visualization Results

Figure 10 extends the analysis of spatio-temporal dynamics. It presents additional heatmap reconstruction examples from the MIBCI dataset, showcasing THVQ-VAE's proficiency in capturing the evolution of spatial brain activity patterns over consecutive time steps. These samples further illustrate its strength in representing dynamic EEG features compared to baseline methods, reinforcing the findings discussed regarding Figure 8 in the main text. Figure 11 presents a multi-domain comparison between the original EEG signal and the reconstructed signal, examining their time-domain waveforms, frequency-domain spectrograms, and spatial heatmaps. The time-domain waveforms show close alignment, indicating effective reconstruction. The frequency-domain spectrograms and spatial heatmaps further reveal that both frequency characteristics and spatial distributions are well-preserved, demonstrating the successful retention of key signal features across all domains.

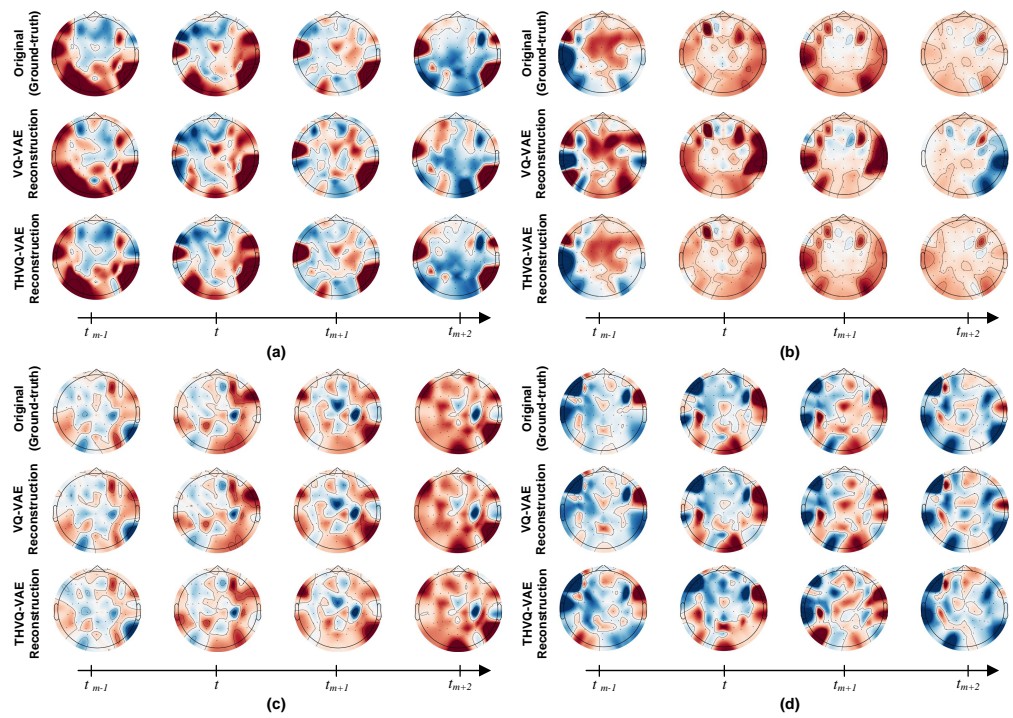

Figure 10: Additional examples from the MIBCI dataset further illustrating THVQ-VAE's enhanced capability in capturing spatio-temporal dynamics of EEG signals.

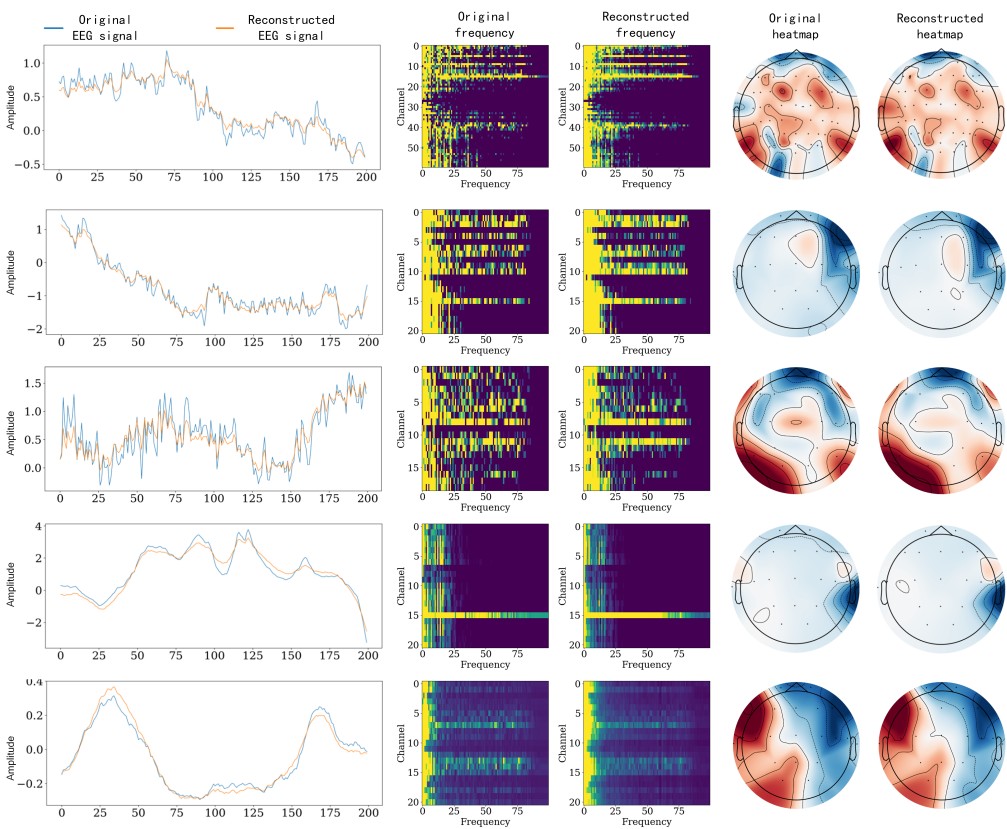

Figure 11: Each row showcases a different EEG channel/segment, comparing: (Left) Original vs. Reconstructed time-domain waveforms; (Center) Original vs. Reconstructed frequency-domain spectrograms (multi-channel); and (Right) Original vs. Reconstructed spatial heatmaps at a specific time point.

# H  Limitations and Future Work

While THD-BAR shows promise, its current multi-modal integration primarily serves instruction-based fine-tuning, with deeper fusion with other physiological signals or richer contextual data remaining an area for expansion. Future work will focus on enhancing THD-BAR's efficiency for real-time BCI applications through model compression and exploring non-autoregressive BAR variants. Expanding the pre-training corpus with more diverse EEG data (varied populations, tasks, noise conditions) and integrating advanced artifact handling will be crucial for improving robustness. Furthermore, building upon our current physiologically-derived BTH and its validated effectiveness, future work will rigorously investigate the impact of alternative topological partitioning methods, such as anatomy-based templates and data-driven clustering approaches. We also aim to extend THD-BAR for deeper multi-modal co-learning with signals like iEEG, fNIRS, fMRI, and behavioral data. Crucially, beyond current classification tasks, future efforts will adapt THD-BAR for a broader range of applications, including EEG denoising, anomaly detection, spatio-temporal localization of neural events or disease biomarkers, and signal reconstruction, to further validate the universality of its learned representations.

