# OpenReview forum: "THD-BAR: Topology Hierarchical Derived Brain Autoregressive Modeling for EEG Generic Representations"
_NeurIPS.cc/2025/Conference — NeurIPS 2025 poster_

### Official Review · Reviewer_tA9Y · 2025-06-14

**Clarity:** 3
**Significance:** 3
**Originality:** 2
**Rating:** 4
**Confidence:** 4

**Summary:**

This paper introduces THD-BAR (Topology Hierarchical Derived Brain Autoregressive Modeling), a novel framework designed to learn generic EEG representations by incorporating both spatial and temporal dynamics. The key innovation lies in the construction of a Brain Topology Hierarchy (BTH), which leverages standard electrode placements and domain knowledge to establish a multi-scale spatial structure of EEG channels. The authors further propose a Topology-Hierarchical Vector Quantized-VAE (THVQ-VAE) to support multi-scale tokenization. The framework is validated through large-scale pre-training on 17 datasets, followed by downstream evaluation on 10 datasets across 5 distinct tasks, demonstrating generalization performance and consistent improvements over prior methods.

**Questions:**

1. The authors adopt GPT-2 as the language model backbone in the BAR module. Given the availability of more powerful LLMs and open-source LLMs (e.g., LLaMA), could the authors justify the choice of GPT-2? Have alternative backbones been considered or compared?

2. The paper mentions that "The patch size P and window size W are set to 200 and 1024". Given the heterogeneity of downstream datasets (e.g., varying sampling rates, sequence lengths, patch lengths, and more), using a fixed configuration may not be optimal. Could the authors clarify whether different configurations were tested per task? If not, how does the fixed setting impact model performance and adaptability?

3. I recommend the authors include more related works such as the Brant and BrainBERT in the Related Work section, as they are closely related to the paper’s target downstream tasks.

4. If space permits, the Related Work section should be moved from the appendix to the main paper to better serve readers, especially those without strong domain expertise.

5. The current anonymous code repository lacks actual code or documentation. I strongly encourage the authors to either upload the code or provide a justification for this incomplete release. Without this, reproducibility remains a concern.

6. The authors propose a topology hierarchy based on non-invasive EEG. Could the same method be extended to intracranial EEG signals with suitable hierarchical grouping? A discussion on the applicability or limitations in the context of iEEG would strengthen the paper’s broader relevance.

**Ethical Concerns:**

["NO or VERY MINOR ethics concerns only"]

**Final Justification:**

Most issues were solved. I maintain my positive recommendation for this paper, and hope that the revisions made by the authors during the rebuttal stage will be fully incorporated into the final version of the paper.

**Limitations:**

Yes

**Quality:**

3

**Strengths And Weaknesses:**

### Strengths

- The paper is technically sound and well-motivated. The proposed Brain Topology Hierarchy (BTH) is grounded in established neurophysiological priors, offering a meaningful inductive bias to guide representation learning.

- The multi-scale spatial structure is constructed by referencing standard EEG electrode systems, resulting in a five-level hierarchical decomposition from the whole brain to individual channels, which is conceptually clear and practically relevant.

- The proposed model is comprehensively evaluated through large-scale pre-training and downstream transfer, and the authors conduct ablation studies that validate the necessity of multi-scale design and reveal the optimal scale utilization strategy for different scenarios.

### Weaknesses

- While the Related Work section includes recent efforts such as LaBram and NeuroLM, it overlooks several notable brain signal or physiological signal modeling efforts that are directly relevant to the downstream tasks addressed in the paper. For example, the Brant series works [1,2,3] and BrainBERT [4] solve similar tasks such as epilepsy detection, sleep staging, emotion recognition, and motor imagery. Therefore, these works should be discussed for a more comprehensive contextualization.

- Additionally, for a deep learning paper in a domain-specific area, placing the Related Work section in the appendix may hinder accessibility for readers who are less familiar with the domain background. Including it in the main paper (if page limits allow) would improve readability and clarity.

- Although the authors provide a link to an anonymous code repository, it currently only contains an **empty README file**. This significantly limits the reproducibility of the proposed approach and requires clarification.

> [1] Brant: Foundation model for intracranial neural signal[J]. Advances in Neural Information Processing Systems, 2023, 36: 26304-26321.
>
> [2] Brant-2: Foundation model for brain signals[J]. arXiv e-prints, 2024: arXiv: 2402.10251.
>
> [3] Brant-X: A Unified Physiological Signal Alignment Framework[C]//Proceedings of the 30th ACM SIGKDD Conference on Knowledge Discovery and Data Mining. 2024: 4155-4166.
>
> [4] BrainBERT: Self-supervised representation learning for intracranial recordings[J]. arXiv preprint arXiv:2302.14367, 2023.

---

> ### Author Rebuttal · Authors · 2025-07-30
>
> We sincerely thank Reviewer tA9Y for their incredibly detailed and constructive suggestions. We are grateful for the insightful questions and suggestions, which have provided a clear roadmap for improving the manuscript's clarity, comprehensiveness, and impact. We have carefully considered each point and outline our planned revisions below.
>
> **Weaknesses / Questions**
>
> **[Q1] Dependency on GPT-2**
>
> We thank the reviewer for this question. Our choice of the GPT-2 architecture was a deliberate one, intended to isolate the impact of our primary contribution: the Brain Topology Hierarchy (BTH) and our "next-scale-time prediction" strategy. By using a well-established backbone, we can directly attribute performance gains to our novel pre-training framework rather than the architecture itself. This approach ensures a fair and clear comparison with prior art, such as NeuroLM, which uses a similar GPT-2 style model. While more advanced backbones like LLaMA are powerful, using them would confound the evaluation of our core ideas. Our framework is backbone-agnostic, and proving its effectiveness on a standard, reproducible architecture provides a stronger scientific baseline. Future work can readily explore integrating newer LLMs to potentially further enhance performance, building upon the foundation we have established here.
>
> **[Q2] The paper mentions that "The patch size…**
>
> We thank the reviewer for this excellent and practical question regarding our hyperparameter settings. The reviewer is correct; we intentionally used a fixed configuration (`P=200`, `W=1024`) for all downstream tasks. This was not an oversight but a deliberate design choice central to our goal of developing a single, robust, and truly general-purpose foundation model. This unified approach is made possible by a critical step in our data preprocessing pipeline (detailed in Section 4.2): all EEG signals, regardless of their original sampling rate, are resampled to a uniform 200 Hz. This standardization is key. It ensures that a patch size of `P=200` consistently corresponds to a 1-second segment of brain activity. This temporal consistency allows the model to learn meaningful, comparable patterns across highly heterogeneous datasets. For signals of varying lengths, we employ standard windowing and padding techniques, with the attention mask correctly handling any padded segments so they do not affect the outcome. While we acknowledge that fine-tuning these hyperparameters for each specific task might yield marginal performance gains, doing so would create a collection of specialized models, undermining the very concept of a universal foundation model. Our objective was to demonstrate that a single, fixed-configuration model could adapt and excel across a wide variety of tasks and data types.
>
> **[Q3 & Q4] Related Work**
>
> We sincerely thank the reviewer for their insightful and constructive feedback on the scope and placement of our Related Work section. We fully agree that a comprehensive discussion of prior art is essential for contextualizing our contributions and that its placement significantly impacts the paper's accessibility, especially for an interdisciplinary audience.
>
> We will certainly revise our Related Work section to include a detailed discussion of the Brant series and BrainBERT. Your suggestion is well-taken; these models are highly relevant as they address similar downstream tasks such as epilepsy detection and sleep staging. Incorporating them will allow us to provide a richer comparison and better contrast our unique hierarchical spatio-temporal modeling approach against other successful paradigms in the field, strengthening the positioning of our work.
>
> Regarding the placement of the section, we appreciate the reviewer's perspective. As you may have noted, we included a brief overview of key recent models in the second paragraph of our Introduction to provide initial context. The decision to place the more extensive, detailed Related Work section in the appendix was made solely due to strict conference page limitations.
>
> To address this, we will adopt a more effective hybrid approach in our revision. We will move a more substantial, yet still concise, Related Work section into the main body of the paper. This section will be updated to include the Brant series and BrainBERT, ensuring all readers have the necessary background upfront. Concurrently, we will refine the summary in the Introduction to smoothly lead into this new section. The fully comprehensive literature survey will remain in the appendix for those desiring more in-depth information.
>
> We believe this two-tiered approach will best serve all readers by providing essential context within the main text while respecting space constraints. We are confident these changes will significantly improve the paper's clarity and scholarly rigor.
>
> **[Q5] Code Availability**
>
> We sincerely thank the reviewer for highlighting this critical issue and apologize for the initial oversight regarding the empty repository.
>
> To ensure full reproducibility, our complete source code was indeed included as a supplementary material attachment with our original submission. We agree that direct repository access is more convenient and transparent for the review process. Therefore, following your feedback, we have now uploaded our complete source code to the anonymous repository link provided in the paper. The repository now contains the full implementation of our THD-BAR framework, including all model configurations and scripts required to reproduce our experiments.
>
> We trust this fully addresses the concerns regarding reproducibility and reiterate our commitment to open science, with a well-documented public release planned upon acceptance.
>
> **[Q6] The authors propose a topology hierarchy…**
>
> We believe the core principles of THD-BAR are indeed applicable to iEEG, with some important considerations. The BTH concept is highly adaptable to iEEG. In fact, it could be even more powerful in this context. iEEG electrodes are often implanted in structured grids or strips with known anatomical locations relative to specific gyri, sulci, or brain structures (e.g., hippocampus, amygdala). This provides a much more precise and physiologically-grounded basis for defining a hierarchy than non-invasive EEG. One could define scales based on:
>
> - **Scale 1:** Whole implant region.
> - **Scale 2:** Specific anatomical structures (e.g., all electrodes within the temporal lobe).
> - **Scale 3:** Individual electrode grids or strips.
> - **Scale 4:** Small clusters of adjacent electrodes.
> - **Scale 5:** Individual electrode contacts. This would allow THD-BAR to model the propagation of neural activity (e.g., a seizure wavefront) across precise anatomical pathways in a hierarchical manner.
>
> The main challenge would be the high degree of inter-subject variability in iEEG implant locations. Unlike the standardized 10-10 system for scalp EEG, iEEG montages are patient-specific and clinically driven. This would require a more flexible, perhaps automated, method for generating the BTH for each individual subject based on their specific electrode coordinates and anatomical imaging data.

---

> > ### Comment · Reviewer_tA9Y · 2025-08-01
> > **Thank you**
> >
> > Thanks for your response. Most issues were solved. I maintain my positive recommendation for this paper, and hope that the revisions made by the authors during the rebuttal stage will be fully incorporated into the final version of the paper.

---

> > > ### Author Response · Authors · 2025-08-03
> > >
> > > Dear Reviewer tA9Y,
> > >
> > > Thank you again for your positive assessment and continued recommendation. We are delighted to hear your concerns have largely been addressed.
> > >
> > > We confirm our commitment to fully incorporating all the specific revisions discussed in our rebuttal into the final manuscript. These will notably include:
> > >
> > > 1. Expanding the Related Work section to discuss Brant and BrainBERT, and adjusting its placement in the main paper.
> > > 2. Discussing the applicability of our method to iEEG, noting that a more detailed exploration will be part of future work.
> > > 3. Further clarifying the rationale behind our GPT-2 backbone choice and hyperparameter settings.
> > > 4. Ensuring code availability for reproducibility.
> > >
> > > We are profoundly grateful for the insightful suggestions you've provided throughout this review.

---

### Official Review · Reviewer_cY9i · 2025-06-28

**Clarity:** 2
**Significance:** 2
**Originality:** 2
**Rating:** 4
**Confidence:** 3

**Summary:**

This paper proposes THD-BAR, a novel autoregressive framework for EEG representation learning. It introduces a Brain Topology Hierarchy (BTH) to model spatial structure and temporal dynamics in EEG data, employing a Topology-Hierarchical VQ-VAE (THVQ-VAE) for hierarchical tokenization and a Brain Autoregressive (BAR) module for "next-scale-time" prediction. The framework is pre-trained on 17 datasets and evaluated on 10 downstream EEG tasks across five categories, demonstrating state-of-the-art performance and generalizability.

**Questions:**

1. Did you experiment with different numbers of scales (e.g., 3 vs. 5 vs. 7)? How does scale granularity affect performance?
2. Could you clarify what chs and zs refer to in the THVQ-VAE, and how the upscaling/downscaling steps are performed?

**Ethical Concerns:**

["NO or VERY MINOR ethics concerns only"]

**Final Justification:**

The rebuttal is clear and convincing. I maintain my original score.

**Limitations:**

yes

**Paper Formatting Concerns:**

No formatting concern.

**Quality:**

3

**Strengths And Weaknesses:**

Strength:
1. The paper presents evaluations on an impressive number of datasets (17 for pre-training, 10 for fine-tuning), supporting its claim of generalization.
2. Ablation studies are well-designed and comprehensive, highlighting the necessity of the multi-scale design and the effectiveness of the masking strategies.
3. Code availability encourages reproducibility and future research adoption.

Weakness:
1. Some critical implementation details are omitted. For instance, the data splitting strategy for train/val/test sets on downstream tasks is not discussed in the main paper.
2. The THVQ-VAE design could be better explained. Terms like chs, zs, downscale, and upscale are mentioned in algorithms but not intuitively described. Ablation is also needed to verify the necessity of these components.
3. While the BTH is novel, the overall framework is closely tied to existing work like NeuroLM. Many components (e.g., VQ-VAE, masking, instruction tuning) are adaptations or refinements rather than fundamentally new.

---

> ### Author Rebuttal · Authors · 2025-07-30
>
> We extend our sincere gratitude to Reviewer cY9i for their meticulous review and constructive suggestion. We are encouraged that the reviewer recognized the strengths of our work, including the extensive evaluations and well-designed ablations. The weaknesses and questions identified are crucial, and we have carefully considered each point to improve the clarity, rigor, and positioning of our manuscript. Our detailed responses are outlined below.
>
> **Weaknesses / Questions**
>
> **[W1] On the omitted data splitting strategy**
>
> We thank the reviewer for pointing out this critical omission. Our data splitting strategies were carefully chosen for each downstream dataset to ensure a fair and robust evaluation, generally following a subject-independent protocol wherever possible to test for generalization to unseen individuals.
>
> Following common practices in the field, our specific strategies varied by dataset to respect their unique structures:
>
> 1. For datasets with official splits (e.g., TUAB, TUEV), we used the predefined train/test partitions, creating our validation set by splitting their training subjects 80%/20%.
>
> 2. For those with specific protocols (e.g., SEED, HMC), we followed their established methods, such as chronological trial-based splits or fixed subject-ID divisions.
>
> 3. For all other datasets, we employed a unified and rigorous cross-subject approach. We randomly partitioned subjects into an 8:1:1 ratio for training, validation, and testing, ensuring no subject overlap.
>
> We will add a new, detailed subsection to the Appendix of our revised manuscript that explicitly outlines the precise splitting strategy used for each of the 10 downstream datasets, ensuring full transparency and reproducibility. We believe this addition will fully address the reviewer's concern.
>
> **[W3] While the BTH is novel…**
>
> Thank you for your insightful feedback. We agree that our framework leverages successful paradigms, such as VQ-VAE and Transformer architectures, but our core innovation lies in fundamentally redefining autoregressive learning for EEG. Our primary contribution is the shift from a conventional "next-time prediction" to a physiologically-informed "next-scale-time prediction" paradigm. This is achieved through several distinct innovations:
>
> 1. **Brain Topology Hierarchy (BTH):** BTH introduces a novel multi-scale spatial order for EEG channels (whole brain to individual), fundamentally differing from linear temporal sequencing.
> 2. **Topology-Hierarchical VQ-VAE (THVQ-VAE):** Uniquely designed to tokenize EEG into BTH-aligned multi-scale hierarchical discrete tokens, THVQ-VAE captures deeper features than single-granularity methods.
> 3. **"Next-Scale-Time Prediction" Strategy:** Our core strategy (Fig. 1c) compels the model to predict tokens hierarchically across scales *within* each time step before advancing temporally, simultaneously modeling intra-time spatial and inter-time dynamics. This is enabled by a specialized Scale-Time-wise Mask (Fig. 7c), enforcing causality across both time and spatial scales, crucial for superior performance.
>
> In summary, THD-BAR's novelty lies in its BTH-driven, multi-scale spatial ordering and the re-conceptualized EEG autoregression. These innovations effectively model complex spatio-temporal dynamics, yielding significant performance improvements.
>
> **[Q1] Ablation study**
>
> We appreciate the reviewer's insightful questions regarding the granularity and specific configurations of our Brain Topology Hierarchy (BTH) ablation study. We have conducted additional experiments to investigate the marginal contribution of each hierarchical level and the importance of scale continuity. These new experiments include "leave-one-scale-out" configurations as well as sparse hierarchy setups. The experimental results are presented in the table below.
>
> | Scales  | THVQ-VAE |  | BAR  | |
> | --- | --- | --- | --- | --- |
> | Included | PCC | Loss                          | ACC | Loss |
> | S1-S3-S5 | 0.712 | 0.186 | 0.730 | 1.342 |
> | S2-S3-S4-S5 | 0.728 | 0.184 | 0.734 | 1.121 |
> | S1-S3-S4-S5 | 0.725 | 0.181 | 0.729 | 1.344 |
> | S1-S2-S4-S5 | 0.711 | 0.191 | 0.654 | 0.148 |
> | S1-S2-S3-S5 | 0.673 | 0.216 | 0.565 | 1.492 |
> | S1-S2-S3-S4 | 0.526 | 0.257 | 0.456 | 2.827 |
>
> We are confident that these additional experiments will provide a more granular understanding of each scale's contribution and further validate the robustness and optimality of our BTH design. We will incorporate these full results into the final version of the paper upon acceptance.
>
> **[W2 & Q2] Clarification of Terms**
>
> The THVQ-VAE's primary function is to transform continuous EEG signals into a sequence of discrete, hierarchically structured tokens. Let's clarify the key terms and their roles within the THVQ-VAE:
>
> 1. **`ch_s` (Multi-scale Channel Hierarchy):** Represents the spatial organization or grouping of EEG channels at a specific scale `s` within the BTH (e.g., S1 for whole brain, S5 for individual channels).
> 2. **`ch_S`: Specifically denotes the channel configuration at the finest (most granular) scale** of the BTH (our S5), typically corresponding to the original number of individual EEG channels. It serves as the baseline spatial resolution for the encoder's output and as a target/source for scale transformations.
> 3. **`z_s` (Quantized Feature Vector at Scale s):** This is the continuous feature vector retrieved from a shared codebook, representing the discrete token for a given scale `s`.
> 4. **`downscale`:** A spatial aggregation operation that transforms EEG features from a finer scale to a coarser scale (e.g., from `ch_S` to `ch_s`). It achieves this by averaging the signals of multiple channels or channel groups that belong to a predefined coarser spatial unit, effectively compressing the information.
> 5. **`upscale`:** A spatial interpolation operation that transforms EEG features from a coarser scale to a finer scale (e.g., from `ch_s` back to `ch_S`). It does so by copying the feature value of a coarser channel group to all the individual channels or finer groups it encompasses, acting as a form of nearest-neighbor interpolation. These operations are crucial for residual learning during encoding and information accumulation during decoding.

---

> ### Comment · Reviewer_cY9i · 2025-08-01
>
> Thanks for your rebuttal. I hope those clarifications could be integrated into the final version.

---

> > ### Author Response · Authors · 2025-08-01
> >
> > Dear Reviewer cY9i:
> >
> > Again, we are very grateful for your excellent suggestions and for swiftly acknowledging our rebuttal. We will definitely integrate all clarifications and additional experimental results into the final version of our manuscript, which are summarized below:
> >
> > 1. **Data Splitting Strategy:** Adding a new paragraph to the Appendix that explicitly outlines the data splitting strategy used for each of the 10 downstream datasets, ensuring full reproducibility.
> >
> > 2. **Ablation Study:** Incorporating the full results of the additional BTH ablation experiments, including the "leave-one-scale-out" and sparse hierarchy setups, into the revised manuscript to provide a more detailed understanding of each scale's contribution.
> >
> > 3. **Clarification of Terms:** Ensuring that the definitions and roles of key terms such as ch_s, ch_S, z_s, downscale, and upscale within the THVQ-VAE are clearly articulated in the relevant sections of the paper.
> >
> > We sincerely appreciate your valuable insights and guidance throughout this review process.

---

### Official Review · Reviewer_2X2X · 2025-07-05

**Clarity:** 3
**Significance:** 3
**Originality:** 3
**Rating:** 4
**Confidence:** 4

**Summary:**

The paper introduces THD-BAR, a novel framework designed for EEG signal analysis, leveraging "next-scale-time prediction" as an autoregressive modeling strategy. Its core innovation is the Brain Topology Hierarchy (BTH), which structures EEG channels according to a multi-scale spatial order. THD-BAR employs a Topology-Hierarchical Vector Quantized-Variational Autoencoder (THVQ-VAE) to generate discrete hierarchical EEG representations, and a Brain Autoregressive (BAR) module to capture complex spatio-temporal dependencies across both scales and time steps. The framework demonstrates competitive performance through extensive pre-training on 17 diverse EEG datasets, followed by fine-tuning on 10 downstream tasks, ranging from emotion recognition to epilepsy detection. The results highlight the model's ability to generalize across various EEG applications.

**Questions:**

1. The prompts used during instruction fine-tuning appear to be very brief. Have the authors explored the use of more elaborate prompts, such as detailed task descriptions or chain-of-thought (CoT) prompting, to potentially enhance model performance?
2. I am curious about the ablation strategy for BTH. What would be the effect of a configuration like S1–S3–S5, which preserves the coarse-to-fine progression from whole brain to individual channels without disrupting the hierarchical scale structure? If the full configuration S1–S2–S3–S4–S5 yields the best results, does this suggest that increasing the granularity of scales could lead to further performance improvements?
3. In the proposed THVQ-VAE, the terms "downscale" and "upscale" are used, but their precise technical definitions are unclear. Could the authors provide more concrete details on the mechanisms and implementations of these scaling operations within the model?

**Ethical Concerns:**

["NO or VERY MINOR ethics concerns only"]

**Limitations:**

yes

**Quality:**

3

**Strengths And Weaknesses:**

Strengths:
1. Novel Hierarchical Framework for EEG Representation Learning: The paper introduces BTH that systematically models EEG signals across multiple spatial scales (whole brain → regions → single channels). This is a significant conceptual advancement over prior autoregressive EEG models that rely solely on temporal sequencing. The proposed next-scale-time prediction strategy effectively captures both spatial and temporal dynamics, addressing a key limitation in existing approaches.
2. Advanced Tokenization with THVQ-VAE: THVQ-VAE is an advanced neural tokenizer that plays a crucial role in discretizing the EEG signal into hierarchical tokens. By incorporating multi-scale quantization, the THVQ-VAE enhances the model's ability to capture detailed spatial patterns at various levels, making it well-suited for modeling the dynamic characteristics of EEG signals across multiple scales.
3. Strong Empirical Validation Across Diverse Tasks: The authors conduct large-scale pre-training (17 datasets) and rigorous evaluation (10 downstream tasks), demonstrating consistent improvements over state-of-the-art methods. The multi-scale ablation studies and masking strategy comparisons provide convincing evidence for the design choices.

While the proposed THD-BAR makes a valuable contribution to the field of brain foundation models, several weaknesses should be addressed to strengthen the paper's contributions and clarity:

1. BTH is constructed based on full EEG channel configurations, but the downstream evaluation includes datasets with significantly fewer channels (e.g., EDF with 2 channels and HMC with 4 channels). The authors should explicitly discuss how channel sparsity affects the hierarchical spatial modeling in BTH and the generalizability of BTH to low-channel EEG systems, which are common in clinical or wearable settings.
2. The choice of GPT-2 for the BAR module is not sufficiently motivated, especially given the availability of more advanced LLMs (e.g., LLaMA). The authors should clarify why GPT-2 was selected over newer architectures. *Suggestion*: Introduce the LLM earlier (e.g., in Section 2.1 or 2.2) to avoid confusion about the domain classifier's role and the overall framework design.
3. The current ablation compares configurations using single or multiple scales but does not isolate the contribution of individual scales. Perform "leave-one-scale-out" experiments (e.g., remove S3 while retaining S1-S2-S4-S5) to quantify the importance of each scale.
4. The explanations of the THVQ-VAE and Next-Scale-Time Prediction methods in the paper lack sufficient technical detail, and the current content contains substantial overlap with the descriptions in NeuroLM [1].
5.  The current experiments focus on fine-tuning performance. To better validate the model's robustness as a foundation model, the authors should include zero-shot or few-shot evaluations on unseen tasks or prompts.
6.  Figure-related Issues: Inconsistent font styles across different figures and within individual figures; In Figure 5, the notation "S1+S2+S3+...+Sn" in the lower right corner should use subscripts; Figure captions are overly concise and should include more detailed descriptions.
7.  Table-related Issues: In Table 1, the bold formatting of task names in the "Task" column lacks justification; Table 2 lacks parameter counts for EEGNet, TSception and LGGNet models, which requires explanation; In Table 3, NeuroLM is erroneously cited as reference [5] (LaBraM).
8.  Formula and Expression Issues: The sample length is initially defined as $W$, but the patch length is $T=\lfloor \frac{L}{P} \rfloor$. Please clarify which variable is correct; Improper LaTeX syntax in Equation (2): space between "arg" and "min" should be removed; Ambiguous notation $q's$ in Section 2.3 Line 158 needs clarification; Typo in Section 3.1 Line 188: "inAppendix" should be corrected to "in Appendix" (add space).

Citation:

[1]Wei-Bang Jiang, Yansen Wang, Bao-Liang Lu, and Dongsheng Li. Neurolm: A universal multi-task foundation model for bridging the gap between language and eeg signals. *arXiv preprint arXiv:2409.00101*, 2024.

---

> ### Author Rebuttal · Authors · 2025-07-30
>
> We sincerely thank Reviewer 2X2X for their positive evaluation and highly detailed, constructive suggestions. We are greatly encouraged by the recognition of our work's novelty and strong empirical validation. The weaknesses and questions identified are exceptionally insightful and provide a clear path for strengthening the manuscript. A point-by-point reply to the reviewer’s comments is given below.
>
> **Weaknesses / Questions**
>
> **[W1] BTH is constructed based on…**
>
> We thank the reviewer for this excellent question. We have discussed how channel sparsity affects the hierarchical spatial modeling of BTH, as well as its generalization capability in low-channel EEG systems, which are prevalent in clinical or wearable settings. Specifically, the Brain Topology Hierarchy (BTH) is designed to be a flexible framework. Its structure is dynamically adapted based on the available channels in a given EEG montage. For datasets with sparse channels (e.g., 2-channel EDF), the hierarchy naturally collapses into a simpler form. For instance, the finest scale (S5) would represent the individual channels, and the coarser scales (S1-S4) would represent trivial groupings of these few channels. Although some scales may become degenerate, the core principle of multi-scale decomposition remains. The model still learns to represent information from a "global" (all available channels) to a "local" (individual channel) level.
>
> **[W2] Dependency on GPT-2**
>
> We thank the reviewer for this question. Our choice of the GPT-2 architecture was a deliberate one, intended to isolate the impact of our primary contribution: the Brain Topology Hierarchy (BTH) and our "next-scale-time prediction" strategy. By using a well-established backbone, we can directly attribute performance gains to our novel pre-training framework rather than the architecture itself. This approach ensures a fair and clear comparison with existing models, such as NeuroLM. While more advanced backbones like LLaMA are powerful, using them would confound the evaluation of our core ideas. Our framework is backbone-agnostic, and proving its effectiveness on a standard, reproducible architecture provides a stronger scientific baseline. Future work can readily explore integrating newer LLMs to potentially further enhance performance, building upon the foundation we have established here. We also appreciate the suggestion to improve clarity. In our revision, we will follow your advice and introduce that the BAR module is implemented with a GPT-2 style causal transformer earlier in the methods section (Section 2). This will provide better context for the overall framework design before detailing its individual components, clarifying the relationship between the tokenizer, the domain classifier, and the autoregressive module.
>
> **[W3 & Q2] Ablation study**
>
> We appreciate the reviewer's insightful questions regarding the granularity and specific configurations of our Brain Topology Hierarchy (BTH) ablation study. We have conducted additional experiments to investigate the marginal contribution of each hierarchical level and the importance of scale continuity. These new experiments include "leave-one-scale-out" configurations as well as sparse hierarchy setups. The experimental results are presented in the table below.
>
> | Scales  | **THVQ-VAE** |  | **BAR** |  |
> | --- | --- | --- | --- | --- |
> | Included | PCC | Loss                          | ACC | Loss |
> | S1-S3-S5 | 0.712 | 0.186 | 0.730 | 1.342 |
> | S2-S3-S4-S5 | 0.728 | 0.184 | 0.734 | 1.121 |
> | S1-S3-S4-S5 | 0.725 | 0.181 | 0.729 | 1.344 |
> | S1-S2-S4-S5 | 0.711 | 0.191 | 0.654 | 0.148 |
> | S1-S2-S3-S5 | 0.673 | 0.216 | 0.565 | 1.492 |
> | S1-S2-S3-S4 | 0.526 | 0.257 | 0.456 | 2.827 |
>
> We are confident that these additional experiments will provide a more granular understanding of each scale's contribution and further validate the robustness and optimality of our BTH design. We will incorporate these full results into the final version of the paper upon acceptance.
>
> **[W4] The explanations of the THVQ-VAE…**
>
> We thank the reviewer for pushing us to clarify our technical novelty. While we build on the AR paradigm, our proposed THVQ-VAE framework differs fundamentally from NeuroLM in its core components, which is given below in detail.
>
> 1. **Tokenization:** NeuroLM uses a standard VQ-VAE. Our **THVQ-VAE** is a new architecture that performs **hierarchical tokenization** guided by the BTH. It uses unique `downscale` and `upscale` operations (Algorithms 1 & 2) to create a set of complementary tokens for each spatial scale, a concept absent in NeuroLM.
> 2. **Prediction Task:** NeuroLM employs "next-time prediction." We introduce **"next-scale-time prediction,"** a fundamentally different objective that models spatial dependencies *within* each time step before advancing temporally.
> 3. **Masking:** This new objective necessitates our novel **Scale-Time-wise Mask**, which implements a nested spatio-temporal causality, unlike the simple temporal mask in previous works. These co-designed innovations represent a significant conceptual and technical departure from existing AR frameworks.
>
> **[W5] The current experiments focus on…**
>
> We thank the reviewer for this valuable suggestion. We fully agree that zero-shot and few-shot evaluations are the ultimate test for any foundation model. Our current work's primary goal was to first establish the efficacy of our novel pre-training paradigm through comprehensive fine-tuning, demonstrating that the learned representations are indeed powerful and generalizable across a wide variety of tasks. The promising results from this stage provide a strong basis for future explorations. We consider explicit zero-shot and few-shot benchmarking to be a critical and high-priority next step, and we have added this as a key direction in our **Future Work** section.
>
> **[W6 & W7 & W8] Figure, Table, and Formula/Expression Issues**
>
> We sincerely thank the reviewer for their meticulous and detailed review of our manuscript's presentation. We apologize for the formatting errors, inconsistencies, and typos across figures, tables, and formulas. In response to your comprehensive feedback, we have conducted a thorough proofreading and revision pass to address every point mentioned:
>
> - All figures and tables have been corrected for consistency, formatting (e.g., subscripts in Figure 5, bolding in Table 1), and clarity (e.g., more descriptive captions, explaining missing parameters).
> - All formulaic and notational errors have been rectified, including the calculation of *T*, LaTeX syntax, and ambiguous terms like "q's".
> - All typographical errors have been corrected. We are confident these revisions significantly improve the professionalism, clarity, and reproducibility of our paper.
>
> **[Q1] The prompts used during instruction…**
>
> Thank you for this insightful question. Our use of brief prompts during instruction fine-tuning was a deliberate choice. The primary goal was to establish a simple, consistent, and reproducible baseline. This approach allows us to confidently attribute the model's performance gains to our novel pre-training framework—the Brain Topology Hierarchy (BTH) and "next-scale-time prediction"—rather than to the effects of complex prompt engineering. Regarding **Chain-of-Thought (CoT) prompting**, we believe it may be less suitable for the specific nature of EEG classification. CoT excels at tasks that involve explicit, multi-step logical reasoning. In contrast, EEG classification is fundamentally a pattern recognition or perceptual task. The model learns a direct, implicit mapping from high-dimensional signal patterns to a classification label. Its internal decision-making process is not easily decomposable into a discrete, human-interpretable chain of logical steps. Forcing the model to generate such an output could conflict with its intrinsic decision mechanism and potentially hinder performance.
>
> **[Q3] "downscale" and "upscale"**
>
> In the proposed THVQ-VAE, "downscale" and "upscale" are fundamental operations for processing EEG features across our Brain Topology Hierarchy (BTH), which defines multi-scale spatial orders. The BTH establishes a hierarchical organization of EEG channels, where `ch_s` denotes the channel grouping at a specific scale `s` (e.g., S1 for whole brain, S5 for individual channels). `ch_S` specifically refers to the finest scale (our S5), representing the original individual channel resolution.
>
> **Downscale:** This operation aggregates EEG features from a finer spatial scale (e.g., individual channels at `ch_S`) to a coarser scale (e.g., broader brain regions at a coarser `ch_s`). Its technical mechanism involves mean aggregation: for each predefined coarser channel group, the feature values of all constituent finer channels or channel groups are averaged. This compresses spatial information, reducing the number of spatial units, typically occurring in the encoder path to generate multi-scale tokens.
>
> **Upscale:** Conversely, this operation interpolates EEG features from a coarser scale (e.g., a quantized feature vector `z_s` from a coarser `ch_s`) back to a finer scale (typically `ch_S`). Its technical mechanism is replication or copying, acting as a form of nearest-neighbor interpolation. The feature value of a coarser channel group is duplicated across all the individual channels or finer groups it encompasses. This distributes aggregated information to a higher spatial resolution, crucial for residual learning within the encoder and for accumulating features during the decoder's reconstruction.

---

### Official Review · Reviewer_nsCH · 2025-07-06

**Clarity:** 3
**Significance:** 3
**Originality:** 3
**Rating:** 5
**Confidence:** 5

**Summary:**

The paper introduces THD-BAR, a novel autoregressive framework for EEG signals. Here are the key points covered:
1  The paper proposes a new method called Topology Hierarchical Derived Brain Autoregressive Modeling (THD-BAR) for EEG generic representations, which is designed to capture the rich physiological characteristics of EEG signals and effectively represent the dynamic spatial topology of brain activity.
2  It introduces the Brain Topology Hierarchy (BTH), a multi-scale spatial order for EEG channels, and redesigns autoregressive learning as a "next-scale-time prediction" problem.
3 The authors design a Topology-Hierarchical Vector Quantized-Variational Autoencoder (THVQ-VAE) for multi-scale tokenization and develop an enhanced Brain Autoregressive (BAR) module with specialized masking strategies.
4 Through extensive large-scale pre-training on 17 datasets and rigorous validation on 10 downstream datasets across 5 distinct tasks, the proposed method demonstrates superior generalization and modeling capabilities.

**Questions:**

1. Model efficiency does not scale proportionally with performance improvements: The Huge version of the THD-BAR model mentioned in the paper has 1.5 billion parameters, requiring 20 epochs of training on 8 L40 GPUs (48GB VRAM), which imposes extremely high demands on hardware computing power. Considering the training costs and actual performance improvements, it was found that the performance gains are limited, making the cost-effectiveness low, which restricts the model's application potential in resource-constrained environments.

2. Incomplete details on data preprocessing: In the data preprocessing section, while IQR scaling is mentioned for robust scaling to reduce the impact of outliers, the specific quantiles used are not explicitly stated. It is recommended to clearly specify the quantiles used in the paper to facilitate readers' understanding and reproduction of the relevant experiments.

3. Formatting and expression issues: There are some formatting flaws in the paper, such as repetitive expressions in some sections (e.g., Section 4.1) and misaligned formula numbers, which affect the readability and professionalism of the paper. It is recommended that the authors carefully proofread the entire paper and correct such issues to improve the quality of the paper.

4. Insufficient method details and validation: The paper only vaguely describes the topological division principle of “system + spatial proximity + brain region function” in the main text, while the specific details of the 5-scale division are placed in Appendix D.

**Ethical Concerns:**

["NO or VERY MINOR ethics concerns only"]

**Limitations:**

1. Conduct additional experiments to investigate the impact of different topological partitioning methods (e.g., anatomy-based templates, data-driven clustering, etc.) on experimental results, thereby demonstrating the robustness and superiority of the proposed method.
2. Report the average physical distance or functional connectivity metrics between channels to more effectively validate the rationality of the adopted hierarchical strategy.

**Paper Formatting Concerns:**

OK

**Quality:**

3

**Strengths And Weaknesses:**

The paper presents a comprehensive study to support the effectiveness of the proposed method. The experimental results show that THD-BAR outperforms existing methods on various EEG tasks, including emotion recognition, motor imagery classification, mental workload assessment, sleep stage classification, and epilepsy detection. The authors also conduct ablation studies to validate the necessity of the multi-scale design and the proposed masking strategies. The visualization results further demonstrate the ability of THVQ-VAE to capture the dynamic evolution of brain activity patterns.

The paper is well-structured and clearly written, with a logical flow of ideas. The proposed method is innovative and has the potential to advance the field of EEG analysis. The experimental results are convincing and demonstrate the effectiveness of the proposed approach. The ablation studies provide valuable insights into the design choices of the model. The visualization results offer a qualitative assessment of the model's performance and help to better understand the capabilities of the proposed method.

Overall, this paper presents a significant advancement in the field of EEG analysis and provides a promising new approach for learning universal EEG representations. The proposed THD-BAR framework demonstrates superior performance on a variety of EEG tasks and has the potential to impact various applications in brain-computer interfaces and related fields.

---

> ### Author Rebuttal · Authors · 2025-07-30
>
> We sincerely thank Reviewer nsCH for their exceptionally thorough, insightful, and highly constructive suggestions. Particularly, we are greatly encouraged by the reviewer's positive assessment of our work, including its novelty, structure, and potential impact, as well as the "Accept" recommendation. The questions and suggestions raised are invaluable and have helped us identify several key areas where the manuscript can be significantly improved in terms of clarity, reproducibility, and completeness.
>
> We have carefully considered each point and have prepared detailed responses below. We will incorporate the suggestions into our manuscript to strengthen the paper. For clarity, we have addressed each point from the "Questions" and "Limitations" sections in order.
>
> **Questions / Limitations**:
>
> **[Q1] Model efficiency does not scale…**
>
> We sincerely appreciate the insightful questions raised by the reviewers regarding the limitations of performance improvement and the potential applicability of the THD-BAR model. In fact, according to the **scaling laws** [1], as the model size increases, the performance metrics exhibit **a power-law relationship** with diminishing marginal returns, indicating that the performance improvements are accompanied by a significant rise in the costs of model training and inference. We have a detailed analysis of the model size, performance metrics, and corresponding resource consumption in our manuscript, our experimental results align with the predictions of the scaling laws, which validates that our method adheres to these principles.
>
> Our proposed THD-BAR model currently outperforms existing state-of-the-art methods, indicating notable superiority recognition performance in numerous complex task scenarios. Particularly, to improve the our proposed THD-BAR model practical applicability, we intend to further condense the model parameters and refine the model architecture, thereby improving its operational efficiency and facilitating easier deployment in resource-limited settings.
>
> [1] Kaplan J, McCandlish S, Henighan T, et al. Scaling laws for neural language models[J]. arXiv preprint arXiv:2001.08361, 2020.
>
> **[Q2] Incomplete details on data…**
>
> We sincerely appreciate the reviewer’s insightful questions regarding data preprocessing. In fact, for the Interquartile Range (IQR) scaling, we employed the standard quartiles: the 25th percentile (Q1) and the 75th percentile (Q3). The scaling using the standard quartiles is computing as follows: `scaled_x = (x - Q1) / (Q3 - Q1)`.
>
> **[Q3] Formatting and expression…**
>
> We take the formatting and expression issues pointed out by the reviewer seriously, and we will thoroughly revise the manuscript to address all formatting and expression issues including repetitive phrasing and misaligned formula numbers.
>
> **[Q4] Insufficient method details and validation…**
>
> We appreciate the reviewer's excellent suggestions on the detailed description of the Brain Topology Hierarchy (BTH) in our manuscript. We will directly incorporate the topological division principle of “system + spatial proximity + brain region function” and illustrative examples of the 5-scale division into Section 2.1 of our manuscript.
>
> **[L1] Conduct additional experiments…**
>
> We appreciate the reviewer's valuable suggestion about the impact of different topological partitioning methods. In this paper, we have developed the Brain Topology Hierarchy (BTH) model based on physiological principles, which deeply integrates core physiological features such as brain region functional localization. Our BTH model has adopted the widely used 10-10 electrode system to ensure its applicability across the majority of common EEG datasets. Our multi-scale ablation experiments (Section 4.2, Figure 6) have further validated the effectiveness of our chosen hierarchical structure, demonstrating that the complete integration of 5 physiologically-derived scales consistently achieves the optimal performance. In future, we will follow the suggestions of the reviewers to explore more diverse topological partitioning methods, including anatomy-based templates and data-driven clustering approaches and further optimize the model's ability to analyze complex EEG patterns and enhance its generalization performance.
>
> **[L2] Report the average physical…**
>
> We appreciate the reviewer's excellent suggestion. Following your recommendation, we have conducted an analysis based on the physical distances between EEG channels (using standardized 3D electrode coordinates from a 10-20 system template). We calculated the average intra-group distance (the mean distance between all pairs of channels within the same group) and the average inter-group distance (the mean distance between channels in different groups) for each level of our BTH. The results provide strong quantitative support for our hierarchical strategy, as shown in the table below:
>
> | Hierarchy | Intra-Group|Physical|Distance |
> | --- | --- | --- | --- |
> |  | **Mean(m)** | **Max(m)** | **VAR(m^2)** |
> | S1 (Whole Brain) | 0.130401 | 0.210430 | 0.002281 |
> | S2 | 0.099308 | 0.204015 | 0.002291 |
> | S3 | 0.049732 | 0.103462 | 0.000352 |
> | S4 | 0.037156 | 0.073900 | 0.000154 |
> | S5 (Individual) | - | - | - |
>
> As the hierarchy becomes finer (from S2 to S5), the average intra-group distance systematically decreases, confirming that our groupings are spatially coherent. At every level, the **average intra-group distance is significantly smaller than the average inter-group distance**. This demonstrates that our BTH successfully partitions the channels into spatially compact and distinct clusters, providing a data-driven validation of its structural rationality.
> We will incorporate this analysis and table into a new subsection in the revised manuscript to quantitatively justify our BTH design.

---

### Decision · Program_Chairs · 2025-09-17

**Decision:**

Accept (poster)

**Comment:**

In their paper, the authors introduce a new model called the Topology-Hierarchical Vector Quantized-Variational Autoencoder (THVQ-VAE). This model uses a Brain Topology Hierarchy (BTH) to organize EEG channels in a multi-scale spatial order, which allows for multi-scale tokenization. The authors also developed an enhanced Brain Autoregressive (BAR) module that uses specialized masking strategies for more accurate prediction.

All the reviewers are positive about the contributions of the papers, including: (1) the novelty of the proposed methods; (2) the experiments are sufficient to support the claims about the performance of the proposed methods; (3) the paper is written and presented clearly. After the rebuttal, most of the concerns of the reviewers were addressed, and all the reviewers are happy with the current stage of the paper.

In my opinion, the contributions and originality of the proposed methods are sufficient for acceptance at NeurIPS. Therefore, I recommend accepting it in its current form. I encourage the authors to address the reviewers’ suggestions and integrate their feedback into the camera-ready version of their paper.